# Field and laboratory assessment of larvicidal activity of tobacco plants and the cigarette butt waste on *Culex pipiens* (Linnaeus, 1758)*, Aedes aegypti* (Linnaeus, 1762) L. and non-target organisms

**Mohamed M. Baz[1,2]\*, Yasser A. El-Sayed[1], Samah Abdel Gawad[3], Mohammed E. Gad[4], Moustafa Ibrahim[5], Mohammed H. Alruhaili[6,7], Hattan S. Gattan[7,8], Reham M. Mostafa[9], Sarita Kumar[10], Ibrahim E. Hussein[11], Abdelfattah Selim[12]\* Abeer M. Alkhaibari[13], Ahlam F. Moharam[14]**

1 Department of Biology, Faculty of Education and Arts, Sohar University, Sohar, Oman, 2 Department of Entomology, Faculty of Science, Benha University, Benha, Egypt, 3 Parasitology Department, Faculty of Veterinary Medicine, Benha University, Egypt, 4 Department of Zoology and Entomology, Faculty of Science, Al Azhar University, Nasr City, Cairo, Egypt, 5 Physics Department, Faculty of Science, Benha University, Benha, Egypt, 6 Department of Clinical Microbiology and Immunology, Faculty of Medicine, King Abdulaziz University, Jeddah, Saudi Arabia, 7 Special Infectious Agents Unit, King Fahad Medical Research Center, King Abdulaziz University, Jeddah, Saudi Arabia, 8 Department of Medical Laboratory Sciences, Faculty of Applied Medical Sciences, King Abdulaziz University, Jeddah, Saudi Arabia, 9 Botany and Microbiology Department, Faculty of Science, Benha University, Benha, Egypt, 10 Department of Zoology, Acharya Narendra Dev College, Delhi University, New Delhi, India, 11 Department of Mathematics, Faculty of Education and Arts, Sohar University, Sohar, Oman, 12 Department of Animal Medicine (Infectious Diseases), College of Veterinary Medicine, Benha University Toukh, Egypt, 13 Department of Biology, Faculty of Science, University of Tabuk, Tabuk, Saudi Arabia, 14 Departments of Parasitology, Faculty of Medicine, Benha University, Benha, Egypt

\* Mohamed.albaz@fsc.bu.edu.eg (MMB); abdelfattah.selim@fvtm.bu.edu.eg (AS)

**Data availability statement:** All relevant data are within the paper.

**Funding:** The author(s) received no specific funding for this work.

## Abstract

Mosquitoes are the primary vectors that transmit diseases affecting humans. Chemical interventions for mosquito management have caused harm to humans, beneficial organisms, and the environment. Hence, researchers are diverting their interest to comparatively safer botanicals. The tobacco plant (*Nicotiana tabacum* L.) is known to contain nicotine and other diverse bioactive phytochemicals with insecticidal properties. The current study aims to repurpose nicotine-containing cigarette butt waste (CBW), a widespread environmental pollutant that poses ecological and economic challenges due to its persistent and toxic chemical components, as friendly insecticides. The study evaluated the methanol and aqueous extracts from tobacco leaves (*Nicotiana tabacum* L.) and cigarette butts (CBs) of various brands against the larvae of *Culex pipiens* and *Aedes aegypti* under field and laboratory conditions and their effects on non-target organisms. Tobacco leaves methanol extracts exhibited higher larvicidal activity (98.0–100%, at 500 ppm) than aqueous extracts (96.0–100%, at 2000 ppm). The methanol extracts of Merit CBs ($LC_{50} = 15.45$ ppm)

**Competing interests:** The authors have declared that no competing interests exist.

demonstrated greatest effectiveness against *Cx. pipiens* larvae, while the methanol extracts of Cleopatra CBs ($LC_{50} = 25.71$ ppm) were found to be more effective against *Ae. aegypti* 24 hours post-treatment. On the other hand, the aqueous extracts of LM cigarette butts ($LC_{50} = 61.22$ and 68.14 ppm) were recorded as most potent against *Cx. pipiens* and *Ae. aegypti* larvae than other CBs. Interestingly, the oviposition activity was higher and positive in CB-treated blackish color cups than that in control water. Field trial data showed 95% and 84% larval reduction with respective Merit CB and tobacco leaves methanol extracts 24 h post-treatment, with persistence for 6 and 3 days, respectively. GC-MS analyses showed a higher number of terpenes, flavonoids, and phenolic compounds in CB extracts. The 3-(1-methyl-2-pyrrolidinyl)-, (S)-, hexadecen-1-ol, phytol, and docosane were the main identified phytochemical compounds. Our findings demonstrate that tobacco and CBs extracts are effective larvicides and represent eco-friendly alternatives for mosquito control.

## 1. Introduction

Tobacco (*Nicotiana tabacum* L.), a widely cultivated crop worldwide belonging to the Solanaceae family is naturally found in regions including Australia, southwest Africa, the Americas, and the South Pacific [1]. It has wide economic and social importance with multifarious uses, including chewing, inhaling, and smoking. High use of tobacco worldwide has led to a dramatic increase in the amount of cigarette butt waste, which is reported to increase from 4.5 trillion units in 2006 to 5.7 trillion in 2016, with expected rise up to 9 trillion units by 2025 [2]. These butts constitute a major environmental source of urban waste contributing to environmental pollution globally incurring high costs. These have been found to contain a complex mixture of chemical compounds, including the highly water-soluble nicotine, a toxic compound [3] along with approximately environmentally persistent 4,000 different compounds like polycyclic aromatic hydrocarbons, heavy metals, arsenic, and tar leading to the release of toxic substances into aquatic environments and posing a threat to aquatic organisms and water sources [4–6].

A typical cigarette butt consists of approximately one-third ash, tobacco fibers, and filters. Several researchers have studied the prevalence and socioeconomic factors of smoking, as well as the amount of nicotine in cigarettes [7,8]. Attempts are being made to recycle these butts and make their beneficial use to decrease the environmental load [5,9]. The use of crude extracts from discarded cigarette butts has also been suggested as a corrosion inhibitor for oil pipelines [10], fiber modifier in asphalt and bitumen, and in the manufacture of lightweight bricks [11,12].

Mosquitoes spread numerous dangerous human and veterinary diseases, including malaria, dengue, yellow fever, filariasis, Japanese encephalitis, chikungunya, and *Streptococcus* genus of cattle [1–4,13,14]. Although no region of the world is free of vector-borne diseases, mosquito-borne illnesses have a global economic impact that includes lost commercial production and employment, the burden of disease and death, poverty, and social debilitation [5,6]. People are hopeful that many of these

chemical compounds, known to kill microbes, pests, and insects, could potentially control mosquito-transmitted diseases [7–10].

For years, residential areas, hospitals, suburbs, and many urban areas, as well as pastures and animal stables, have used pesticides, chemical pesticides, and fungicides to control mosquitoes, insect pests and control insect-borne diseases. Despite the many advantages of these pesticides, such as their easy access and rapid and effective action, their extensive and indiscriminate use has led to widespread environmental pollution. Excessive use has also harmed non-target species, leading to the development of pest resistance to these pesticides, necessitating their repeated, unregulated application at high doses. The use of organic pesticides, or biopesticides, which are less expensive, more environmentally friendly, and more sustainable, has become necessary due to the risks associated with synthetic pesticides. Use of plants (exudates, essential oils, bark, root, and leaf extracts) and microbes (metabolites, for example) as biopesticides can be a good alternate to synthetic pesticides [11,12,15]. Unlike synthetic pesticides, biopesticides are environmentally sustainable, durable, easily available, and specific in their action. In addition to having a variety of phytochemical components that enable them to act in different ways, botanical pesticides are safer for human use and do not contribute to the production of greenhouse gases [16]. Therefore, many researchers and environmentalists are reattempting the use and recycle plant waste extracts to formulate pesticides in order to utilize them effectively and beneficially instead of discarding them [17].

Hence, keeping in mind the polluting effects of toxic cigarette butt wastes and challenges associated with the chemical interventions in mosquito control programs, the present study aimed to explore tobacco leaves and cigarette butt wastes as natural, sustainable and environmentally safe alternatives for mosquito control [18]. We believe that exploiting cigarette but waste this would not only provide a safer option of mosquito control but would also represent an innovative solution to reduce the volume of this pollutant. Tobacco leaves contain several alkaloids, including the extremely poisonous alkaloid nicotine, one of the deadliest plant products in its pure form. Furthermore, nicotine is the most important in the chemical profile of tobacco, as it constitutes 90–95% of the total alkaloids; nornicotine and anabasine are present to a lesser extent. Total alkaloid concentrations range from 25 to 38 mg/g of dry matter, with variations depending on the cultivar and leaf location. In addition to alkaloids, the leaves contain flavonoids such as lutein, apigenin, and rutin (300–800 µg/g of dry matter) and terpenes, including diterpenes and sesquiterpenes [19,20], all of which contribute to the plant's insecticidal activity through their toxic effects and diverse chemical compositions, making tobacco a promising natural source of biopesticides.

A few studies have reported the use of cigarette butt waste as an unusual way to get rid of pests [21]. The study reported the larvicidal potential of various plant and industrial waste extracts against *Cx. pipiens*, with cigarette filters causing 100% mortality while Lipton tea killing 97% of them. In addition, cigarette filters and apricot kernel extracts were more attractive substrates for egg-laying by female adults. Likewise, alkaloids, phenols, sesquiterpenoids, and terpenoids derived from cigarette butt extracts were excellent at killing *Zea mays* L. pests [22].

Turning cigarette butt waste (CBW) into a biopesticide can be an alternative control tool against insecticide-resistant mosquito vectors. According to Munteanu and Didilescu [23], the primary constituents of tobacco are tar, carbon monoxide, and the alkaloid nicotine, with varying concentrations found in different sections of the plant. Many researchers showed that excellent insect control and complete biodegradability of nicotine make it an effective pesticide component [24]. Alkaloids have been observed to be one of the potent compounds against larvae of *Ae. albopictus* [25] Many alkaloids can kill insects at low concentrations by inhibiting nerve impulses from traveling through the synaptic pathway *via* acetylcholinesterase (AChE) activity. As a result, the mode of action on insect vectors varies depending on the structure of their molecules [21]. According to Simon-Oke, Afolabi [26], alkaloids function by narrowing blood vessels and lowering autonomic nervous system activity, which makes insecticides more effective in killing mosquito larvae and tampering with the mosquito's life cycle.

We believe that CBW would possess high efficacy against *Cx. pipiens* and *Ae. aegypti* that are resistant to common pesticides since tobacco extracts are effective against many insect vectors, including mosquitoes [27]. In the present

work, we hypothesized that bioactive phytochemicals present in the cigarette butt waste would have lethal effects on the larvae of *Cx. pipiens* and *Ae. aegypti* mosquitoes, and as compared with *Nicotiana tabacum* leaf extracts, it can act as an alternative green insecticide. We also evaluated cigarette butt and *N. tabacum* leaf extracts against non-target organisms. The research paper flow chart summarizes the work steps as shown in Fig 1.

## 2. Materials and methods

### 2.1 Plant materials and analysis

**2.1.1 *Nicotiana tabacum* collection.** The protocol of work was approved by the Ethics Committee of the Faculty of Science, Benha University (Code: BUFS-REC-2025–355 Ent). The study was conducted in accordance with the local legislation and institutional requirements. In this study, no permits were required for this study, as the fieldwork was conducted in areas that do not require special access or authorization. The work complied with all relevant local and national regulations. Tobacco (*Nicotiana tabacum*) leaves were collected from the St. Catherine area, South Sinai, Egypt (33°55 E and 28°30 N), during October and November 2023 (Fig 2). The tobacco leaves collected from the plant were

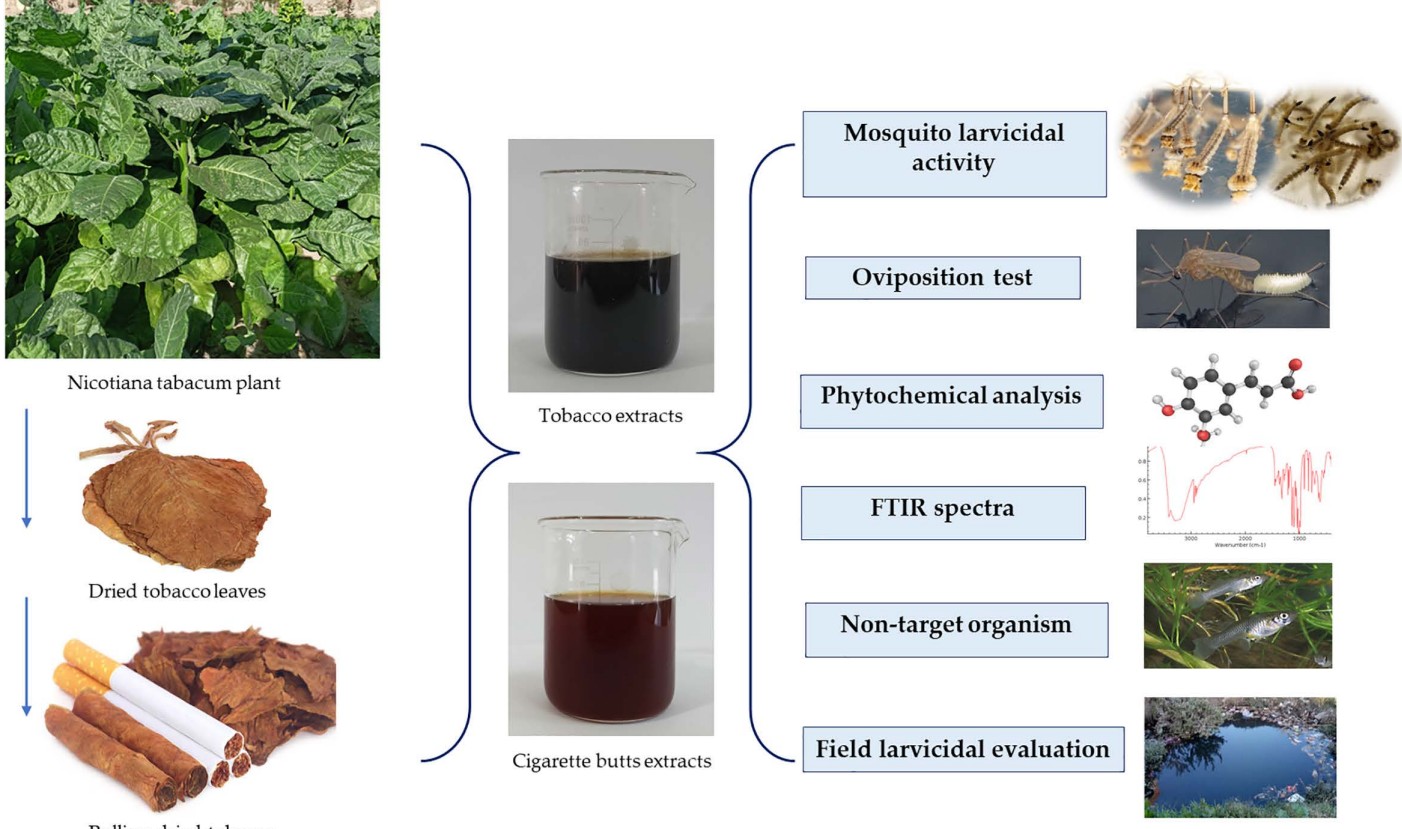

**Fig 1. The figure depicts the research structure.** The preparation of tobacco leaf extracts (TEs) and cigarette butt extracts (CBs) involves drying tobacco leaves and unsmoked tobacco cigarette butts, then extracting them using methanol and aqueous solvents. The crude extracts were evaluated against *Culex pipiens* and *Aedes aegypti* mosquito larvae for their larvicidal activity after 24 and 48 h of exposure to different concentrations of each TE and CB and effects on oviposition by adult females. The metabolic components of the TEs and CBs were identified and characterized using Fourier transform infrared spectroscopy (FTIR), and by screening for the presence of different types of secondary metabolites, determination of total phenolics and total flavonoids, and identification of metabolites by gas chromatography-mass spectrometry (GC-MS). The extracts were also tested for killing mosquito larvae and against non-target organisms to determine their efficacy in the field.

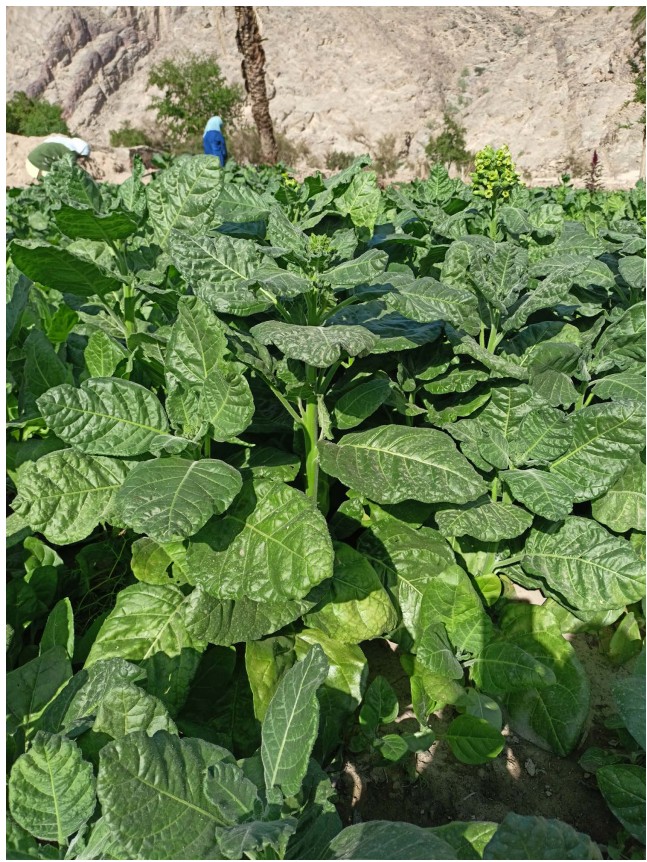

**Fig 2. Tobacco plant, *Nicotiana tabacum* (St. Catherine, South Sina, Egypt).**

identified and authenticated by Dr. Trease Labib, a plant taxonomy consultant at the Egyptian Ministry of Agriculture. A voucher specimen was placed under the code (PHG-P-NT-533) at the herbarium of the Pharmacognosy Department, Faculty of Pharmacy, Ain Shams University. The plant leaves were shaded and air-dried away from sunlight at room temperature (28°C±2, 65% RH) until the plant was well dried and the dry weight was achieved. The dried leaves were ground in a stainless-steel electric blender and transferred to sealed dark containers to protect them from moisture and light [28].

**2.1.2 *Nicotiana tabacum* extraction.** To make the methanol extract, 40 g of dry plant materials was soaked in 100 mL of methanol in a 250 mL stoppered conical flask. The flask was then left to stand at room temperature (27±2°C) for three days, stirring often, until the soluble matter dissolved. The solution was filtered through Whatman No. 1 filter paper using a Büchner funnel. For aqueous extract, 10 g of plant powder was extracted using warm water (60°C) at room temperature. After 48 hours, the extraction was completed, and the solution was filtered through a Büchner funnel with Whatman No. 1 filter paper. A rotary evaporator was used to concentrate the extracts and evaporate the solvent, which were then kept in black bottles for storage [25].

**2.1.3 Cigarette butt collections.** Cigarettes of four brands (Merit, LM, Cleopatra, and Boston) were purchased from authorized shops, while discarded cigarette butts were collected from some tourist cafeterias and households in Benha area, Qalyubiya, Egypt. The amount of tobacco remaining in cigarette butts varied according to length or the remaining part of the smoked cigarette. The length of the entire cigarette, the filter paper, the full tobacco-containing part, and the

tobacco-remaining parts (unsmoked) were measured. Also, their weights with and without cigarette paper were measured to estimate the remaining tobacco (Table 1, Fig 3).

**2.1.4 Cigarette butt extractions.** Merit, LM, Cleopatra, and Boston butts with 2–5 cm of unburnt tobacco was used as experimental Cigarette butts (CBs). To obtain the different test solutions, we adopted a slight modification of the procedures reported previously [27,29]. After crushing, the filter and cigarette papers were removed. The 6 g of CBs (~10 cigarette tobacco) was soaked in 100 mL of deionized water in a 250-mL round bottle flask filled with a methanol solvent, while another 6 g of CBs was soaked in 100 mL of deionized water kept in a water bath (35°C). The CBs solution was left to disintegrate and release chemical leachates for 48 h. After the extraction process was completed, filter paper (Whatman No. 1) and vacuum filtration were used to separate the extract and filter cake. Throughout the filtration process, we cleaned the filter paper twice with more with 10 milliliters of ethanol each time to eliminate any remaining nicotine. To produce viscous crude extracts, a vacuum evaporator set to 100°C and 45 rpm for two hours was used to remove the solvent from the filtrate [24]. Fig 1 shows a schematic depiction of the complete procedure.

## 2.2 Phytochemical analyses

**2.2.1 GC/MS analysis.** The bioactive constituents present in the tobacco leaf methanol and water extracts were identified using Thermo Scientific Trace GC Ultra/ISQ Single Quadrupole MS and TG-5MS fused 0.1 mm, 0.251 mm, and 30 m thick silica capillary columns. An electronic ionizer with ionization energy of 70 eV was used for the process. Helium gas was used as a carrier gas (flow rate: 1 ml/min). Both the injector and the MS transmission line were adjusted at 280°C. Following a two-minute wait, the oven was preheated to 35°C. From there, it was raised to 150°C at a pace of 7°C per minute, 270°C at a rate of 5°C per minute, and finally, 310°C at a rate of 3.5°C per minute (continued for

**Table 1. Weight of cigarettes, tobacco, filters, and leftover tobacco in various cigarettes and butts.**

| Type | Weight of cigarette | Weight of cigarette tobacco | Weight of cigarette filter | Weight of used cigarette filter | Weight of 10 used filter CBs | Weight of 10 cigarettes | Weight of 10 unburned tobacco CBs | Weight of 10 cigarette tobacco |
|---|---|---|---|---|---|---|---|---|
| Cleopatra | 0.875 | 0.75 | 0.125 | 0.127 | 1.27 | 8.75 | 3.30 | 7.50 |
| Boston | 0.811 | 0.68 | 0.131 | 0.142 | 1.42 | 8.11 | 3.16 | 6.80 |
| Merit | 0.741 | 0.60 | 0.141 | 0.160 | 1.6 | 7.41 | 2.81 | 6.00 |
| LM | 0.752 | 0.61 | 0.142 | 0.159 | 1.59 | 7.52 | 2.78 | 6.10 |

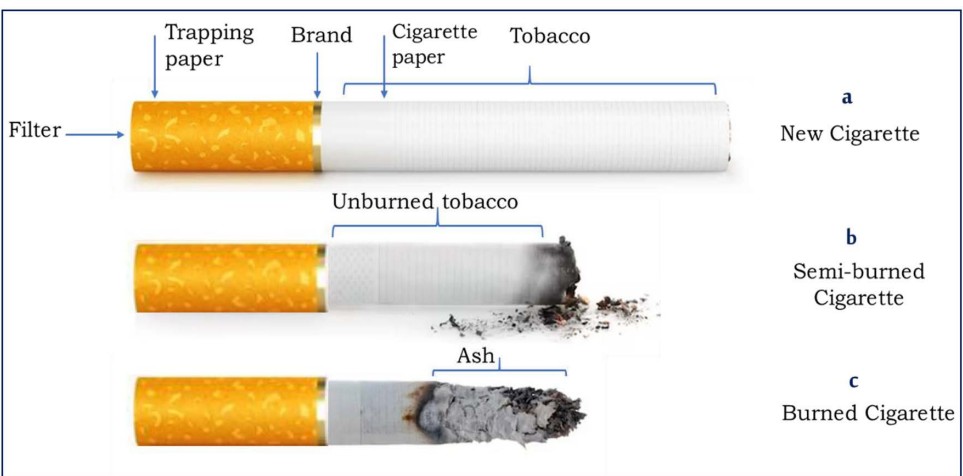

**Fig 3. Cigarette components with possible lengths of cigarette butts containing tobacco residue.**

10 min). All components found were explored for quantification using a relative peak area. By matching their retention durations and mass spectra with information from the NIST and Willy libraries on the GC/MS equipment, the kind of chemicals were determined in the extracts. The aggregate spectrum of user-generated reference libraries was used for identification. Single-ion chromatographic reconstructions were used to assess peak uniformity. Whenever possible, co-chromatographic analysis of reference chemicals was utilized to confirm GC retention durations [30].

**2.2.2 UV-visible spectroscopy.** Tobacco plant leaf and Merit cigarette butt extracts were investigated using double beam UV-visible spectroscopy in the range of 190–900 nm (Edinburgh Instruments DS5, United Kingdom) using a quartz cuvette. Absorbance as a function of wavelengths was plotted to examine the visual effect of tobacco and cigarette butt extracts on adult mosquito oviposition.

**2.2.3 Fourier transform infrared (FTIR) spectroscopy.** Tobacco plant leaf and Merit cigarette butt extracts were examined using FTIR spectroscopy (Bruker ALPHA II) in the range of 4000−400 cm$^{-1}$. The resulting spectral information was used to identify the function groups present in the tobacco and cigarette butt extracts.

## 2.3 Mosquito larvicidal assay

**2.3.1 Rearing of mosquito larvae.** *Cx. pipiens* and *Ae. aegypti* strains used in this experiment were obtained from a colony maintained at 28°C ± 2, 75 ± 5% humidity, and 13:11 h (L/D) photoperiod in the Medical Entomology Department, Faculty of Science, Benha University. Larvae were reared in metal trays (12 cm in diameter and 2 cm in depth) containing 100 larvae per tray and filled with dechlorinated water. Larvae were fed on Tetramin fish food mixed with ground bread powder in a ratio of 1:3. Tetramin is rich in proteins, vitamins, and minerals that support the growth of mosquito larvae. Every two days, water was added to the larval rearing enamel plates and changed if necessary. The pupae formed were transferred from the metal trays to a beaker filled with dechlorinated water and placed in screened cages (size 35 × 35 × 40 cm), where adults eventually emerged. The hamster mouse periodically supplied blood to female mosquitoes, while the adult mosquito colony received 10% sugar solution [31]. The research title, overall objective of the study, and ethical approval for the use of hamsters in this study were presented to the Ethics Committee of the Faculty of Science, and it is approved under Code (BUFS-REC-2025–355 Ent).

**2.3.2 Larvicidal activity of tobacco plant extract.** The methanol and aqueous extracts of *N. tabacum* leaves were evaluated against the 3rd larval instar of *Cx. pipiens* and *Ae. aegypti* under laboratory conditions. The 3$^{rd}$ larval instar was treated with the following concentrations: 25, 50, 100, 200, and 500 ppm (1 g/1L) for methanol extracts and 100, 250, 500, 1000, and 2000 ppm for aqueous extracts. Twenty larvae per concentration were transferred to a glass beaker containing 250 mL of distilled water. Five replicates were used for each concentration. Mortalities were recorded 24 and 48 h after the initial exposure [32] and post-treatments (PT).

**2.3.3 Larvicidal activity of Cigarette butt extracts.** Cigarette butt methanol and aqueous extracts of Merit, LM, Cleopatra, and Boston Cigarette brands were evaluated against the 3$^{rd}$ larval instar of *Cx. pipiens* and *Ae. aegypti* under laboratory conditions. The 3$^{rd}$ larval instar was treated with the following concentrations: 3, 6, 12, 25, and 50 ppm (1 g/1L) for methanol extracts and 25, 50, 100, 250, and 500 ppm for aqueous extracts. Twenty larvae per concentration were transferred to a glass beaker containing 250 mL of distilled water. Five replicates were used for each concentration. Mortalities were recorded 24 and 48 h after the initial exposure [32] and post-treatments (PT).

**2.3.4 Oviposition bioassays.** To perform oviposition tests, 50 females and 50 males aged 3–4 days were placed in screened cages (35 × 35 × 40 cm in size) and fed with 10% sucrose solution for 3 days. After a 24-hour starvation period, female mosquitoes were fed on blood meals from hamsters fixed in the middle of the cage for 1 hour, and then the blood source was removed from the cage to allow the engorged females to digest the blood meals for egg-laying. Glass cups (9 cm deep, 5 cm bottom diameter, 7.5 cm top diameter) were used as egg-laying substrates. The treatment group consisted of three cups filled with cigarette butt extract at varying concentrations (5, 10, and 15 ppm) (Fig 4), and a fourth cup filled with tap water, each holding 200 ml of the solutions. Each of the two extracts of imported and local cigarettes

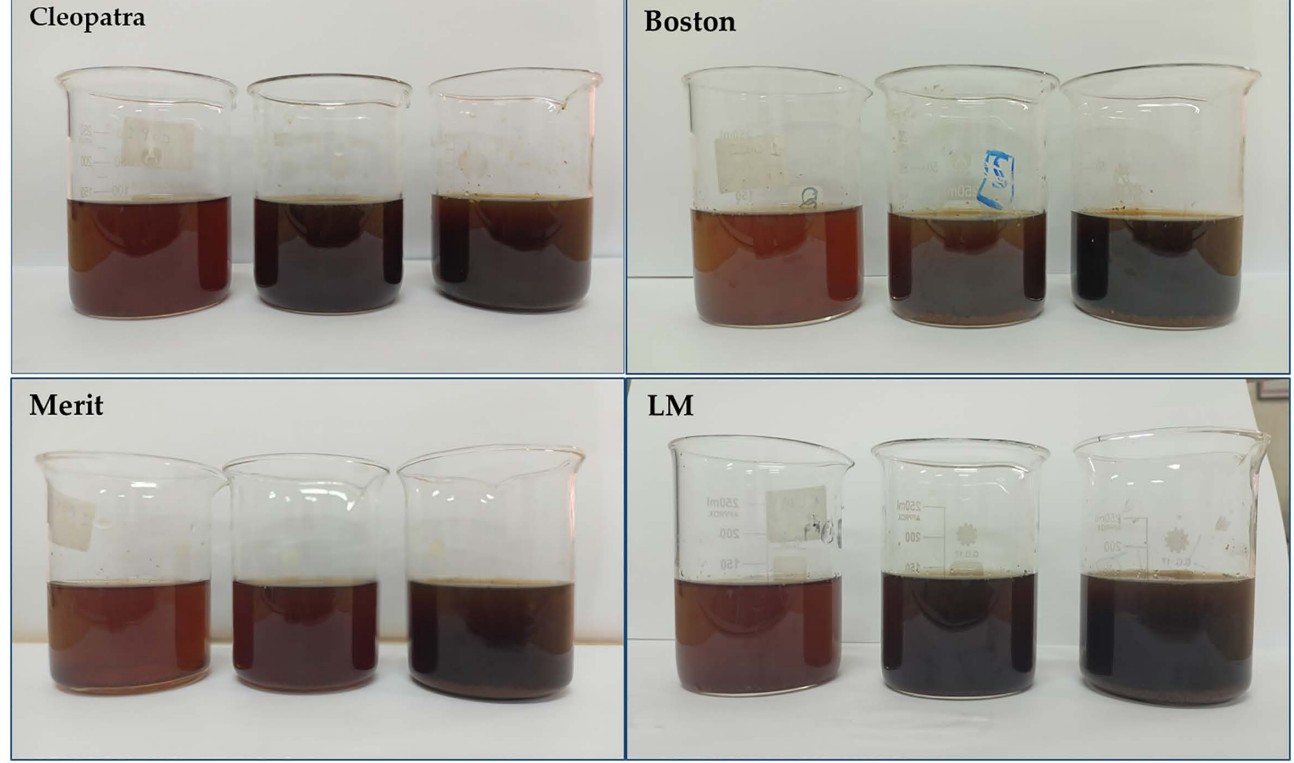

**Fig 4. Cigarette butt extract concentrations to perform oviposition tests.**

(Cleopatra and Merit) (Boston and LM) was placed together. Individual cups were spaced evenly apart inside the cage. The experiment was repeated three times. After 3–5 days from the start of the experiment, eggs were collected and examined under a dissecting microscope to monitor the number of egg deposition [27].

**2.3.5 Activity of tobacco extracts against non-target predators.** Tobacco and cigarette butt extracts were evaluated for their efficacy on prevalent predators in natural mosquito larval habitats as non-target organisms. The effectiveness of tobacco leaves and cigarette butt extracts was evaluated against *Gambusia affinis*, *Cybister* sp., and *Crocothemis erythraea*, which were captured using aquatic traps in various stagnant mosquito larval environments in Kafr Saad village, Qalyubiya, Egypt. Predators were identified and classified by Dr. Yasser El-Sayed at the Entomology Department of Benha University, Egypt, after being captured alive, placed in plastic bags half filled with water, and transferred to the laboratory for categorization and predation assessments [12]. The aquatic predators were kept in separate stock plastic tanks that were half filled with dechlorinated water for 3–5 days before evaluation.

**2.3.6 Field assessment against mosquito larvae.** Nature mosquito water pools with average lengths of 5.25, 3.0, and 0.45 m were chosen in Kafr Saad village, Qalyubiya Governorate, Egypt, for the evaluation of larvicidal efficacy of tobacco leaves and cigarette butt extracts. The selection of the pools for larval raising was based on their high mosquito stage density and the abundance of mosquito breeding sites. An $LC_{95}$ x 2 doses of both tobacco leaves and cigarette butt methanol extracts was applied to the breeding sites. For every treatment, three replicates were conducted. For each larvicidal treatment, mosquito larvae were collected in field water from each site using an enamel pad (450 mL) [32] for counting and comparing them with untreated sites.

## 2.4 Statistical analysis

The data was analyzed by the software SPSS V23 (IBM, USA). The probit analysis was carried out to compute the lethal concentration (LC) values and the two-way analysis of variance (ANOVA) (Post Hoc/Turkey's HSD test). The significant levels were set at $P < 0.05$.

## 3. Results

### 3.1 Efficacy of *Nicotiana tabacum* leaves extracts against mosquito larvae

This study evaluated extracts of tobacco leaves (*Nicotiana tabacum*) on 3rd instar larvae of *Cx. pipiens* and *Ae. aegypti*. After different intervals of exposure, tobacco leaf extracts in this study showed high insecticidal activity against mosquito larvae. The results showed that methanol extracts of tobacco leaves had more toxic effects against mosquito larvae than aqueous extracts. The mortality percentage (MO%) using 500 ppm tobacco leaves methanol extracts was 100% for *Cx. pipiens*, with a $LC_{50}$ (50%, median lethal concentration) of 97.09 and 67.64 ppm, and 98% and 100% for *Ae. aegypti*, with a $LC_{50}$ of 108.75 and 77.87 ppm, respectively, at 24 and 48 h post-treatment (PT) (Table 2, 3). However, the MO% using 2000 ppm aqueous extracts for *Cx. pipiens* and *Ae. aegypti* were 100% and 96% at 24 h (Table 2), with $LC_{50}$ values of 469.60 and 557.31 ppm, and 100% at 48 h PT, with $LC_{50}$ values of 300.80 and 376.03 ppm (Table 2). Tables 2, 3 and Fig 5a showed that the $LC_{50}$ values of *N. tabacum* leaves methanol extract (67.64 and 77.87 ppm) were more effective than aqueous extracts ($LC_{50} = 300.80$ and 376.03 ppm), against *Cx. pipiens* and *Ae. aegypti,* 48 h PT.

### 3.2 Efficacy of cigarette butt extracts against mosquito larvae

The assessment of different brands of cigarette butts against mosquito larvae showed significant activity of methanol and aqueous extracts, the efficacy varying depending on the type of cigarette butt. The mortality percent (MO%) of *Cx. pipiens* larvae imparted by 100 ppm cigarette butt methanol extracts was 90, 85, 100, and 100% by Cleopatra, Boston, Merit, and LM brands, respectively (Table 4) and that of *Ae. aegypti* was 90, 89, 88, and 85%, respectively 24 h PT. The corresponding larval mortalities reached 100, 95, 100, and 100% for *Cx. pipiens* and 100, 98, 95, 94 for *Ae. aegypti*, respectively, at 48 h PT (Table 4). The $LC_{50}$ values of Cleopatra, Boston, Merit, and LM CBs were 24.52, 32.21, 15.45, and 18.46 ppm for *Cx. pipiens* and 25.71, 28.81, 31.01, and 32.78 ppm for *Ae. aegypti,* 24 h post-treatments (Table 5).

On the other hand, MO% of 3rd larval stage of *Cx. pipiens* with 500 ppm of Cleopatra, Boston, Merit, and LM butt aqueous extracts were 99, 95, 100, and 100%, respectively, 24 h PT, and the corresponding values were 100, 98, 100, and 100%, respectively, 48 h PT. Meanwhile, MO% of 3rd larval stage of *Ae. aegypti* following treatment with Cleopatra,

**Table 2. Efficacy of *Nicotiana tabacum* methanol leaf extracts against *Culex pipiens* and *Aedes aegypti* 24 and 48 h post-treatment.**

| Extract | Time (h) | Insect | Concentration (ppm) | | | | | |
|---|---|---|---|---|---|---|---|---|
| | | | 0 | 25 | 50 | 100 | 200 | 500 |
| Methanol | 24 | *Cx. pipiens* | 0±0[aF] | 7.0±1.22[aE] | 19.0±2.92[aD] | 46.0±1.87[aC] | 82.0±2.55[aB] | 100.0±0.00[aA] |
| | | *Ae. aegypti* | 0±0[aF] | 5.0±1.58[bE] | 16.0±1.00[bD] | 46.0±2.92[bC] | 75.0±1.58[bB] | 98.0±2.00[bA] |
| | 48 | *Cx. pipiens* | 0±0[aF] | 12.0±1.22[aE] | 30.0±2.74[aD] | 64.0±1.87[aC] | 98.0±1.22[aB] | 100.0±0.00[aA] |
| | | *Ae. aegypti* | 0±0[aF] | 8.0±1.22[bE] | 25.0±1.58[bD] | 60.0±1.58[bC] | 92.0±1.22[bB] | 100.0±0.00[aA] |
| Aqueous | 24 | *Cx. pipiens* | 0±0[aF] | 9.0±1.87[aE] | 20.0±3.54[aD] | 48.0±3.39[aC] | 80.0±2.24[aB] | 100.0±0.00[aA] |
| | | *Ae. aegypti* | 0±0[aF] | 6.0±1.87[bE] | 16.0±2.92[bD] | 41.0±1.87[bC] | 72.0±2.55[bB] | 96.0±2.45[bA] |
| | 48 | *Cx. pipiens* | 0±0[aF] | 17.0±1.22[aE] | 33.0±2.00[aD] | 65.0±4.18[aC] | 96.0±2.45[aB] | 100.0±0.00[aA] |
| | | *Ae. aegypti* | 0±0[aF] | 11.0±1.87[bE] | 25.0±2.24[bD] | 59.0±1.87[bC] | 88.0±3.39[bB] | 100.0±0.00[aA] |

a, b, & c: There is no significant difference (P>0.05) between any two means for each time within the same column has the same superscript letter. A, B & C: There is no significant difference (P>0.05) between any two means within the same row having the same superscript letter.

**Table 3. Lethal concentrations (ppm) of *Nicotiana tabacum* methanol and aqueous leaf extracts against *Culex pipiens* and *Aedes aegypti*, 24 and 48 h post-treatment.**

| Extract | Time (h) | Insect | LC$_{50}$ (Low-Up.) | LC$_{90}$ (Low-Up.) | LC$_{95}$ (Low-Up.) | Slope±SE | Chi (Sig.) |
|---------|----------|--------|---------------------|---------------------|---------------------|----------|------------|
| Methanol | 24 | *Cx. pipiens* | 97.09 (86.75-108.80) | 267.13 (225.63-330.69) | 355.89 (291.87-459.37) | 2.915±0.215 | 4.833 (0.184) |
| | | *Ae. aegypti* | 108.75 (96.96-122.39) | 313.72 (262.63-392.92) | 423.61 (343.78-554.32) | 2.785±0.205 | 1.264 (0.737) |
| | 48 | *Cx. pipiens* | 67.64 (60.82-75.08) | 168.85 (145.95-202.64) | 218.81 (184.36-272.43) | 3.227±0.239 | 7.446 (0.058) |
| | | *Ae. aegypti* | 77.87 (69.94-86.69) | 196.99 (168.70-239.73) | 256.25 (213.57-324.26) | 3.180±0.241 | 2.265 (0.519) |
| Aqueous | 24 | *Cx. pipiens* | 469.60 (413.50-532.17) | 1557.43 (1291.31-1974.13) | 2187.79 (1751.43-2914.72) | 2.4614±0.182 | 6.895 (0.075) |
| | | *Ae. aegypti* | 557.31 (492.64-630.69) | 1776.90 (1470.70-2261.49) | 2468.37 (1972.99-3301.03) | 2.5451±0.191 | 7.024 (0.071) |
| | 48 | *Cx. pipiens* | 300.80 (168.76-469.27 | 922.04 (724.96-2392.63) | 1266.64 (1055.77-3941.14) | 2.6345±0.199 | 14.138 (0.002) |
| | | *Ae. aegypti* | 376.03 (251.25-1106.28) | 1128.52 (867.82-2231.16) | 1541.02 (1194.64-3448.52) | 2.6852±0.198 | 9.463 (0.023) |

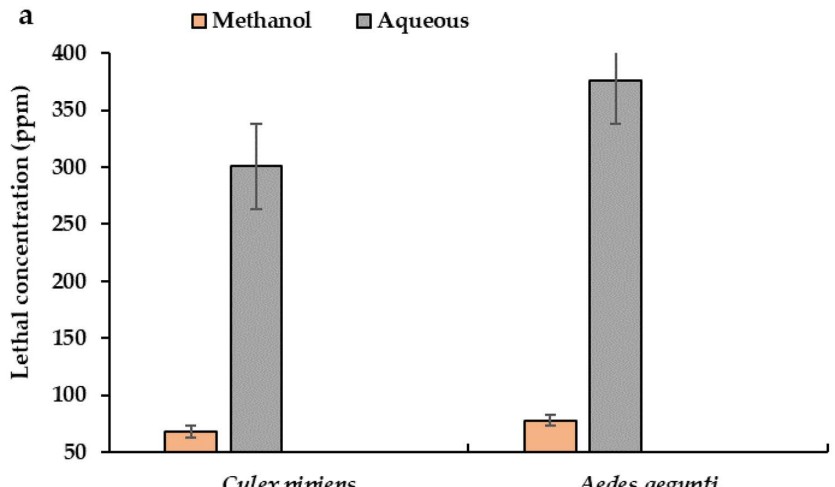

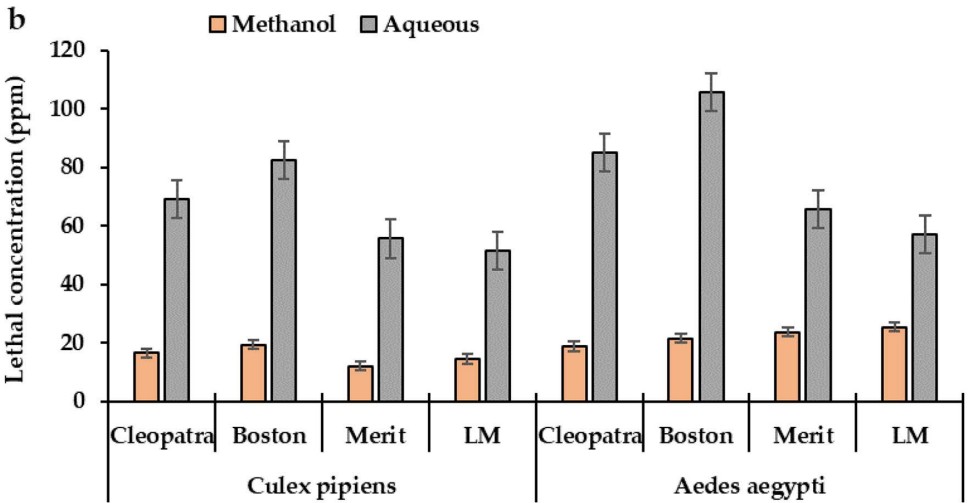

**Fig 5. The mean number of larval mortalities induced by *Nicotiana tabacum* leaves (a) and cigarette but extracts (b) against 3rd larval instars of *Culex pipiens* and *Aedes aegypti*, 48 h post-treatment.**

**Table 4. Efficacy of cigarette butt methanol extracts against *Culex pipiens* and *Aedes aegypti* 24 and 48 h post-treatment.**

| Time | Insect | Type | Concentration (ppm) | | | | | |
|---|---|---|---|---|---|---|---|---|
| | | | 0 | 5 | 10 | 25 | 50 | 100 |
| 24 | *Cx. pipiens* | Cleopatra | 0±0[aF] | 7.00±1.22[cE] | 19.00±2.92[cD] | 50.00±3.54[cC] | 77.00±2.55[cB] | 90.00±1.58[bA] |
| | | Boston | 0±0[aF] | 4.00±1.00[dE] | 14.00±1.87[dD] | 40.00±2.74[dC] | 67.00±2.00[dB] | 85.00±2.24[cA] |
| | | Merit | 0±0[aF] | 14.00±1.87[aE] | 29.00±1.87[aD] | 62.00±3.39[aC] | 94.00±1.87[aB] | 100.00±0.0[aA] |
| | | LM | 0±0[aF] | 10.00±1.58[bE] | 24.00±1.87[bD] | 55.00±2.74[bC] | 88.00±4.06[bB] | 100.00±0.0[aA] |
| | *Ae. aegypti* | Cleopatra | 0±0[aF] | 7.00±1.22[aE] | 19.00±1.87[aD] | 48.00±2.55[aC] | 73.00±1.22[aB] | 90.00±2.74[aA] |
| | | Boston | 0±0[aF] | 5.00±1.58[bE] | 15.00±1.58[bD] | 44.00±1.87[bC] | 70.00±1.58[bB] | 89.00±2.92[abA] |
| | | Merit | 0±0[aF] | 4.00±2.92[bC] | 14.00±1.87[bcD] | 41.00±2.45[cC] | 67.00±3.39[cB] | 88.00±2.00[bA] |
| | | LM | 0±0[aF] | 4.00±1.00[bE] | 13.00±2.55[cD] | 42.00±2.55[cC] | 64.00±4.00[dB] | 85.00±1.58[cA] |
| 48 | *Cx. pipiens* | Cleopatra | 0±0[aF] | 12.00±1.22[cE] | 30.00±2.24[cD] | 60.00±2.74[cC] | 90.00±3.16[cB] | 100.00±0.00[aA] |
| | | Boston | 0±0[aF] | 10.00±0.00[dE] | 26.00±1.87[dD] | 56.00±4.00[dC] | 88.00±2.55[dB] | 95.00±2.24[bA] |
| | | Merit | 0±0[aF] | 19.00±1.87[aE] | 44.00±3.32[aC] | 70.00±2.24[aB] | 100.00±0.00[aA] | 100.00±0.00[aA] |
| | | LM | 0±0[aF] | 15.00±2.24[bE] | 35.00±4.18[bD] | 63.00±2.00[bC] | 95.00±5.00[bB] | 100.00±0.00[aA] |
| | *Ae. aegypti* | Cleopatra | 0±0[aF] | 10.00±1.58[aE] | 23.00±1.22[aD] | 54.00±2.92[aC] | 84.00±2.92[aB] | 100.00±0.00[aA] |
| | | Boston | 0±0[aF] | 9.00±1.87[aE] | 22.00±2.55[aD] | 50.00±1.58[bC] | 80.00±1.58[bB] | 98.00±2.00[bA] |
| | | Merit | 0±0[aF] | 9.00±2.92[aE] | 20.00±2.24[bD] | 46.00±2.92[cC] | 76.00±4.00[cB] | 95.00±2.24[cA] |
| | | LM | 0±0[aF] | 7.00±3.39[bE] | 17.00±3.74[cD] | 46.00±5.10[cC] | 74.00±1.87[dB] | 94.00±1.87[cA] |

[a], [b], & [c]: There is no significant difference (P > 0.05) between any two means for each insect for each period, within the same column have the same superscript letter; A, B & C: There is no significant difference (P > 0.05) between any two means, within the same row have the same superscript letter.

**Table 5. Lethal concentrations (ppm) of cigarette butt methanol extracts against *Culex pipiens* and *Aedes aegypti*, 24 and 48 h post-treatment.**

| Time | Insect | Type | LC$_{50}$ (Low-Up.) | LC$_{90}$ (Low-Up.) | LC$_{95}$ (Low-Up.) | Slope±SE | Chi (Sig.) |
|---|---|---|---|---|---|---|---|
| 24 | *Cx. pipiens* | Cleopatra | 24.52 (21.3-28.20) | 95.05 (76.49-125.41) | 139.54 (107.79-195.10) | 2.178±0.163 | 0.380 (0.944) |
| | | Boston | 32.21 (27.99-37.26) | 127.37 (100.48-173.31) | 188.09 (141.97-272.47) | 2.146±0.166 | 0.096 (0.992) |
| | | Merit | 15.45 (10.49-21.90) | 47.65 (36.02-90.21) | 65.56 (49.54-138.99) | 2.620±0.190 | 8.449 (0.037) |
| | | LM | 18.46 (16.26-20.88) | 57.07 (47.95-70.81) | 78.59 (64.05-101.81) | 2.614±0.185 | 7.325 (0.062) |
| | *Ae. aegypti* | Cleopatra | 25.71 (22.32-29.65) | 103.18 (82.30-137.95) | 152.99 (116.90-217.37) | 2.123±0.161 | 0.099 (0.991) |
| | | Boston | 28.81 (25.13-33.13) | 108.80 (87.26-144.40) | 158.57 (122.08-222.97) | 2.221±0.167 | 0.096 (0.032) |
| | | Merit | 31.01 (27.05-35.69) | 116.48 (93.10-155.45) | 169.50 (129.99-239.82) | 2.229±0.170 | 0.130 (0.987) |
| | | LM | 32.78 (28.47-37.98) | 130.84 (102.89-178.89) | 193.70 (145.65-282.24) | 2.132±0.166 | 0.249 (0.969) |
| 48 | *Cx. pipiens* | Cleopatra | 16.38 (14.37-18.58) | 52.45 (43.92-65.42) | 72.96 (59.18-95.23) | 2.535±0.183 | 5.870 (0.118) |
| | | Boston | 19.31 (16.75-22.19) | 73.05 (59.21-95.50) | 106.52 (82.94-147.49) | 2.217±0.169 | 4.228 (0238) |
| | | Merit | 12.82 (9.90-14.95) | 50.91 (39.97-70.70) | 75.27 (56.19-112.89) | 2.139±0.198 | 5.946 (0.114) |
| | | LM | 14.16 (9.49-20.61) | 45.60 (34.38-89.77) | 63.28 (47.88-140.89) | 2.553±0.188 | 8.835 (0.031) |
| | *Ae. aegypti* | Cleopatra | 19.33 (17.00-21.93) | 62.27 (52.00-77.091) | 86.74 (70.16-113.57) | 2.523±0.180 | 7.246 (0.064) |
| | | Boston | 21.44 (18.83-24.33) | 74.93 (62.44-93.78) | 106.83 (86.14-139.98) | 2.358±0.160 | 3.693 (0.296) |
| | | Merit | 23.64 (20.61-27.10) | 88.57 (71.83-115.57) | 128.80 (100.44-177.62) | 2.234±0.165 | 3.785 (0.285) |
| | | LM | 25.31 (22.14-28.95) | 91.02 (74.14-118.05) | 130.84 (102.63-178.94) | 2.305±0.169 | 2.021 (0.567) |

Boston, Merit, and LM CB aqueous extracts were 95, 90, 98, and 100% at 24 h PT, and 99, 92, 100, and 100% at 48 h PT, respectively (Table 6).

The results in Table 5 and Fig 5b showed that the LC$_{50}$ values of the methanol extracts of Merit (12.82), LM (14.16), Cleopatra (16.38), and Boston (19.31) cigarette butts were lower than those of the aqueous extracts (LC$_{50}$ = 55.45, 51.35, 69.15, and 82.49 ppm, respectively) against the third larval stage of *Cx. pipiens*, 48 hours PT (Table 4, Fig 5b). Similarly,

**Table 6. Efficacy of cigarette butt aqueous extracts against *Culex pipiens* and *Aedes aegypti* 24 and 48 h post-treatment.**

| Time (h) | Insect | Type | Concentration (ppm) | | | | | |
|---|---|---|---|---|---|---|---|---|
| | | | 0 | 25 | 50 | 100 | 250 | 500 |
| 24 | *Cx. pipiens* | Cleopatra | 0 ± 0[aF] | 12 ± 1.22[cE] | 29 ± 1.87[cD] | 55 ± 2.24[cC] | 83 ± 2.55[cB] | 99 ± 1.00[bA] |
| | | Boston | 0 ± 0[aF] | 8 ± 1.22[dE] | 23 ± 1.22[dD] | 52 ± 2.55[dC] | 80 ± 2.24[dB] | 95 ± 1.58[cA] |
| | | Merit | 0 ± 0[aF] | 16 ± 1.87[bE] | 35 ± 2.24[bD] | 65 ± 1.58[bC] | 92 ± 2.00[bB] | 100 ± 0.00[aA] |
| | | LM | 0 ± 0[aF] | 18 ± 1.22[aE] | 39 ± 2.92[aD] | 70 ± 1.58[aC] | 95 ± 2.24[aB] | 100 ± 0.00[aA] |
| | *Ae. aegypti* | Cleopatra | 0 ± 0[aF] | 8 ± 1.22[bE] | 22 ± 1.22[cD] | 47 ± 3.39[cC] | 80 ± 1.58[cB] | 95 ± 2.24[cA] |
| | | Boston | 0 ± 0[aF] | 6 ± 1.87[cE] | 19 ± 1.00[dD] | 46 ± 1.00[dC] | 78 ± 2.00[dB] | 90 ± 1.58[dA] |
| | | Merit | 0 ± 0[aF] | 14 ± 2.92[aE] | 33 ± 2.00[bD] | 62 ± 3.00[bC] | 88 ± 3.39[bB] | 98 ± 2.00[bA] |
| | | LM | 0 ± 0[aF] | 15 ± 1.58[aE] | 35 ± 3.54[aD] | 66 ± 1.87[aC] | 91 ± 3.32[aB] | 100 ± 0.00[aA] |
| 48 | *Cx. pipiens* | Cleopatra | 0 ± 0[aF] | 15 ± 2.74[cE] | 35 ± 2.24[cD] | 65 ± 2.24[cC] | 90 ± 4.18[cB] | 100 ± 0.00[aA] |
| | | Boston | 0 ± 0[aF] | 12 ± 3.39[dE] | 28 ± 2.00[dD] | 60 ± 2.74[dC] | 85 ± 3.16[dB] | 98 ± 1.22[bA] |
| | | Merit | 0 ± 0[aF] | 20 ± 2.24[bE] | 41 ± 2.45[bD] | 74 ± 1.87[bC] | 98 ± 2.55[bB] | 100 ± 0.00[aA] |
| | | LM | 0 ± 0[aF] | 21 ± 2.45[aE] | 44 ± 3.32[aD] | 77 ± 2.55[aC] | 100 ± 0.00[aB] | 100 ± 0.00[aA] |
| | *Ae. aegypti* | Cleopatra | 0 ± 0[aF] | 11 ± 2.45[cE] | 25 ± 3.87[cD] | 51 ± 5.57[cC] | 85 ± 4.47[cB] | 99 ± 4.58[bA] |
| | | Boston | 0 ± 0[aF] | 9 ± 1.87[dE] | 22 ± 3.39[dD] | 49 ± 2.92[dC] | 81 ± 2.92[dB] | 92 ± 2.55[cA] |
| | | Merit | 0 ± 0[aF] | 16 ± 1.87[bE] | 36 ± 4.00[bD] | 70 ± 2.24[bC] | 92 ± 3.74[bB] | 100 ± 0.00[aA] |
| | | LM | 0 ± 0[aF] | 18 ± 3.39[aE] | 38 ± 5.39[aD] | 74 ± 3.32[aC] | 96 ± 2.45[aB] | 100 ± 0.00[aA] |

[a], [b], & [c]: There is no significant difference (P > 0.05) between any two means for each insect, within the same column have the same superscript letter. A, B & C: There is no significant difference (P > 0.05) between any two means, within the same row have the same superscript letter.

against the third larval stage of *Ae. aegypti*, the $LC_{50}$ values of methanol extracts of Cleopatra (19.33), Merit (23.64), Boston (21.44), and LM (25.31) cigarette butts were lower than those of the aqueous extracts ($LC_{50}$ = 90.02, 65.47, 105.52, and 58.61 ppm, respectively), 48 h post-treatments (Table 6, Fig 5b). The aqueous extract of LM butt ($LC_{50}$ = 51.35 and 58.61 ppm) was found to be most effective than rest of the butt extracts against both the third larval stage of *Cx. pipiens* and *Ae. aegypti*, 48 h post-treatments (Table 7, Fig 5b).

**Table 7. Lethal concentrations (ppm) of cigarette butt aqueous extracts against *Culex pipiens* and *Aedes aegypti*, 24 and 48 h post-treatment.**

| Time (h) | Insect | Type | $LC_{50}$ (Low-Up.) | $LC_{90}$ (Low-Up.) | $LC_{95}$ (Low-Up.) | Slope±SE | Chi (Sig.) |
|---|---|---|---|---|---|---|---|
| 24 | *Cx. pipiens* | Cleopatra | 85.40 (74.67-97.45) | 30015 (246.46-385.22) | 428.63 (339.23-579.67) | 2.347±0.174 | 3.192 (0.362) |
| | | Boston | 111.76 (95.51-131.51) | 509.93 (387.14-738.89) | 784.12 (563.79-1230.64) | 1.944±0.168 | 6.910 (0.074) |
| | | Merit | 67.45 (59.20-76.50) | 214.81 (178.99-270.86) | 298.30 (240.16-395.31) | 2.547±0.196 | 2.042 (0.563) |
| | | LM | 61.22 (53.63-69.40) | 194.81 (163.13-243.93) | 270.48 (218.93-355.79) | 2.549±0.197 | 1.229 (0.746) |
| | *Ae. aegypti* | Cleopatra | 106.29 (93.26-121.03) | 382.91 (315.51-487.11) | 550.66 (438.09-735.45) | 2.302±0.159 | 0.168 (0.982) |
| | | Boston | 118.72 (103.55-136.43) | 447.71 (360.35-590.34) | 652.24 (504.46-909.65) | 2.2230.164 | 0.969 (0.808) |
| | | Merit | 74.85 (65.27-85.45) | 264.09 (217.17-338.62) | 377.54 (299.05-510.85) | 2.340±0.178 | 0.386 (0.943 |
| | | LM | 68.14 (59.82-77.28) | 217.02 (180.82-273.63) | 301.37 (242.63-399.30) | 2.547±0.195 | 1.877 (0.598) |
| 48 | *Cx. pipiens* | Cleopatra | 69.15 (60.61-78.52) | 224.99 (187.00-284.55) | 314.35 (252.30-418.12) | 2.501±0.181 | 2.167 (0.538) |
| | | Boston | 82.49 (72.16-94.04) | 286.24 (235.59-366.25) | 407.29 (323.22-548.86) | 2.371±0.177 | 1.503 (0.681) |
| | | Merit | 55.54 (48.95-62.60) | 159.41 (134.30-198.86) | 214.95 (175.23-281.55) | 2.798±0.230 | 2.539 (0.468) |
| | | LM | 51.35 (45.97-58.33) | 140.42 (118.99-172.21) | 186.15 (152.78-242.27) | 2.967±0.251 | 4.247 (0.235) |
| | *Ae. aegypti* | Cleopatra | 90.02 (79.17-102.26) | 295.73 (244.91-375.92) | 414.31 (331.63-551.14) | 2.481±0.181 | 3.268 (0.352) |
| | | Boston | 105.52 (91.85-121.29) | 406.50 (327.37-535.74) | 595.80 (460.70-831.61) | 2.188±0.163 | 0.595 (0.897) |
| | | Merit | 65.47 (57.26-74.48) | 213.03 (176.05-272.39) | 297.65 (237.10-401.60) | 2.501±0.201 | 3.105 (0.375) |
| | | LM | 58.61 (51.77-65.94) | 167.75 (142.80-205.14) | 226.00 (186.85-288.34) | 2.806±0.210 | 2.068 (0.558) |

## 3.3 Oviposition responses of females

Investigations on the oviposition and hatchability rate of eggs laid by female mosquitoes in the CBs solution showed that female mosquito oviposition responses varied significantly with the concentration of CB solutions (CBSO). Results showed that females tended to prefer oviposition in cups containing CBs to cups containing only water. Table 8 and Fig 6 summarize the detailed number of laid eggs and the hatching rate. The comparison of the oviposition cups containing only water to those containing CBs showed lower egg deposition in all CB-tested cups. The mean number of eggs laid in oviposition cups containing 5, 10, and 15 ppm CB solutions was higher than in the control oviposition cups. The number of eggs oviposited was 220, 580, 205, and 611 for Cleopatra, Merit, Boston, and LM solutions at 5 ppm CB solution, and 1146, 1640, 780, and 1500 in 10 CB solutions, respectively. The results showed that egg deposition varied significantly with the amount of CBs solutions (F = 11.056, df = 3, P < 0.001). The OAI values were higher in CBSO colors. Conversely, the egg hatching rate was higher in the water than in the 10 ppm CBs solutions by 11.7, 3.7, 13.8, and 4.9%, respectively.

## 3.4 Efficacy of tobacco extracts against non-target predators

The efficacy of LC$_{50}$ values of *N. tabacum* leaves and CB extracts against several non-target insect predators, *G. affinis, Cybister* spp.*,* and *C. erythraea*, was evaluated. There was a negative significant difference (P = 0.001, df = 2, F = 26.08) in the rate of mosquitofish predation. The rate of predation of *G. affinis* dropped by 53.7% and 71.3%, compared to 82.0% in the untreated groups, respectively. However, there was no significant difference (P > 0.05) in the predation rate of *Cybister* spp. (63.7% and 60.3%) and *C. erythraea* (28.3% and 27.0) on mosquito larvae compared to the untreated groups (61.3 and 25%), respectively (Fig 7).

**Table 8. Efficacy of cigarette butt extract concentration on oviposition rates of female *Culex pipiens*.**

| Parameter | CBs | Concentration | | | |
|---|---|---|---|---|---|
| | | Water | 5CB | 10CB | 15CB |
| Number of eggs rafts oviposited (mean±SE) | Cleopatra | 175.0±33.29[aC] | 220.0±43.59[bC] | 633.3±60.09[cB] | 1146.0±26[cA] |
| | Merit | 175.0±33.29[aD] | 580.0±15.28[aC] | 1383.0±69.29[aB] | 1640.0±66.58[aA] |
| | Boston | 158.3±21.67[aC] | 205.0±18.93[bC] | 415.0±29.30[dB] | 780.0±17.32[dA] |
| | LM | 158.3±21.67[aD] | 611.6±53.72[aC] | 1232.3±97.84[bB] | 1500.0±11.55[bA] |
| Number of eggs laid/ female | Cleopatra | 8.05[aC] | 10.12[aC] | 29.13[bcB] | 52.71[aA] |
| | Merit | 4.63[aD] | 15.35[aC] | 36.61[aB] | 43.41[bA] |
| | Boston | 10.16[aC] | 13.16[aC] | 26.63[cB] | 50.05[aA] |
| | LM | 4.52[aD] | 17.46[aC] | 35.19[abB] | 42.83[bA] |
| Hatching % (mean±SE) | Cleopatra | 97.8±17.76[aA] | 62.1±5.25[aB] | 37.2±4.33[aC] | 11.7±0.41[aD] |
| | Merit | 97.8±17.76[aA] | 40.5±8.43[bB] | 15.1±3.36[bC] | 3.7±0.12[aD] |
| | Boston | 94.4±5.56[aA] | 66.8±10.74[aB] | 39.4±3.88[aC] | 13.8±0.25[aD] |
| | LM | 94.4±5.56[aA] | 41.7±5.98[bB] | 16.9±2.82[bC] | 4.9±0.12[aD] |
| OAI | Cleopatra | 0[aB] | 0.11[bB] | 0.57[bA] | 0.74[aA] |
| | Merit | 0[aC] | 0.54[aB] | 0.78[aA] | 0.81[aA] |
| | Boston | 0[aD] | 0.13[bC] | 0.45[bB] | 0.66[aA] |
| | LM | 0[aC] | 0.59[aB] | 0.77[aAB] | 0.81[aA] |
| F-Value | | Cleopatra (111.76) | Merit (176.63) | Boston (161.53) | LM (112.42) |
| P-Value | | Cleopatra (0.00) | Merit (0.00) | Boston (0.00) | LM (0.000) |

Higher OAI implies a preference for the cigarette butt cup colors.

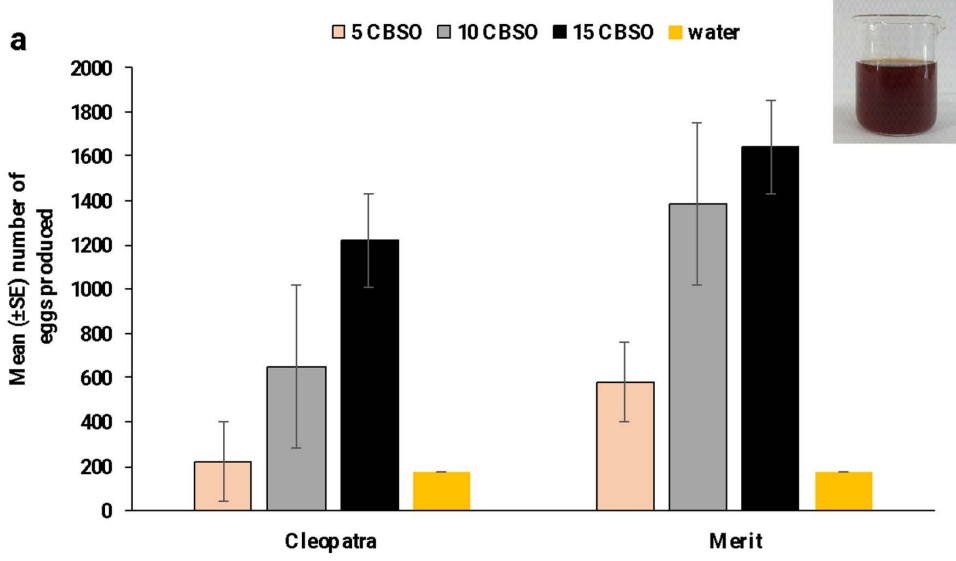

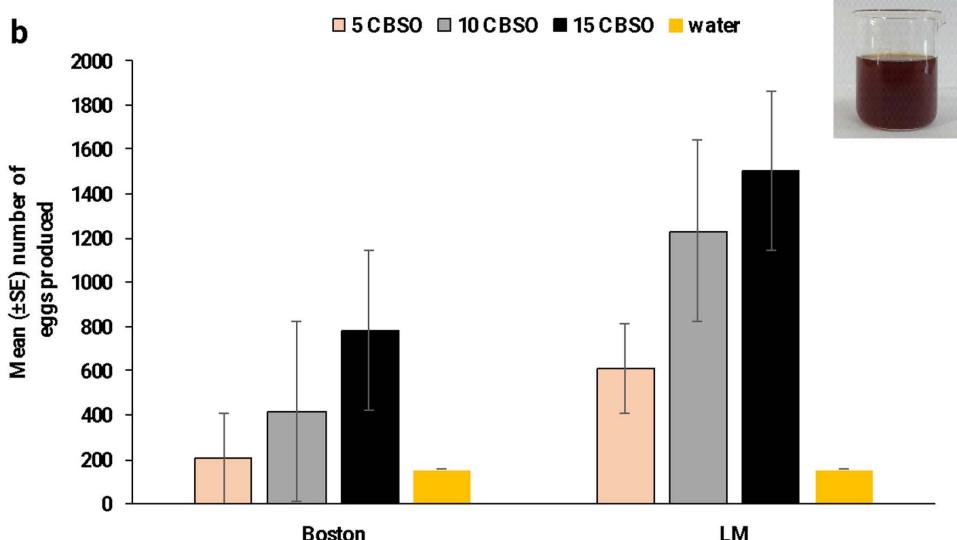

**Fig 6. Impact of cigarette butt cup colors on female oviposition.**

### 3.5 Larvicidal field assessment

The efficacy of *N. tabacum* leaves and Merit CB methanol extracts evaluated under field conditions showed a decreased larval density of 84.0% with *N. tabacum* leaf extract after 24 h post-treatment which continued for three days (Fig 8). On the other hand, the Merit CB extract resulted in a significant reduction of 95.0% after 24 h post-treatment which persisted for six days (Fig 8).

### 3.6 Phytochemical analyses

**3.6.1 Optical measurements.** In order to emphasize the main reason for mosquito mortality, it was required to test the effect of the visual appearance of the tobacco or cigarette butt samples on mosquitoes. Optical measurements included the absorption and emission of light in tested samples, which give the samples their apparent color. Absorption

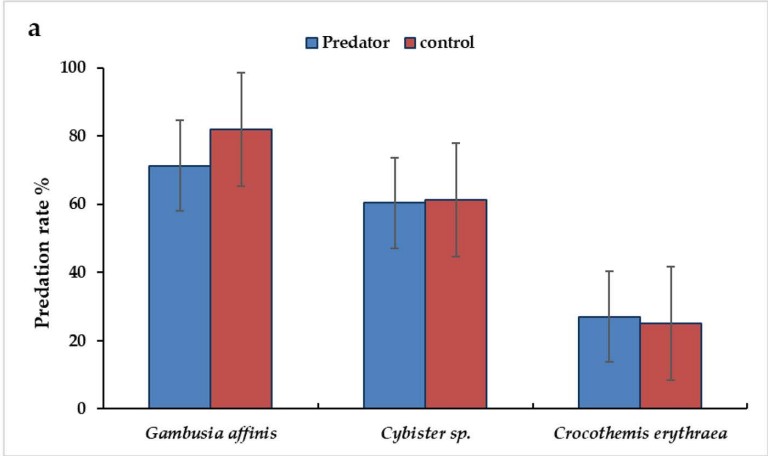

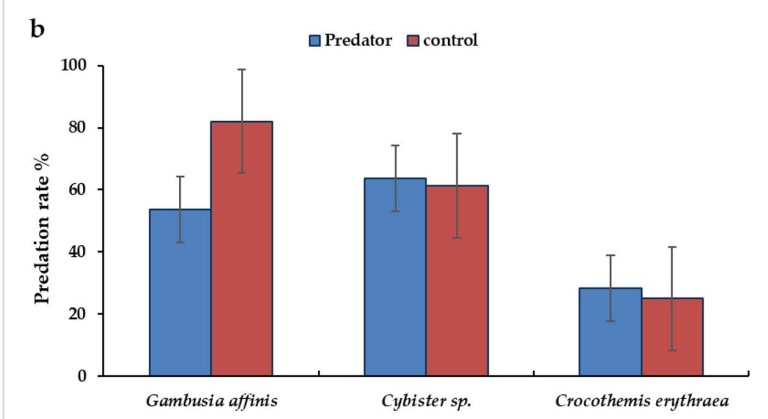

**Fig 7. Impact of *Nicotiana tabacum* leaves (a) and Merit cigarette butt (b) methanol extracts against non-target predators.**

of the tobacco and Merit cigarette but extracts were comparable (Fig 9). Both samples showed peaks in the UV region 200–350 nm, but the tobacco sample showed an extra peak at 603 nm.

**3.6.2 FTIR measurements.** The FTIR spectra of the tobacco leaf extract and the Merit cigarette butt extract showed similar characteristic absorption bands, indicating the presence of multiple functional groups responsible for their bioactivity (Fig 10). Both samples exhibited a broad absorption peak around 3326 cm$^{-1}$, corresponding to O–H stretching vibrations typical of hydroxyl groups found in phenolic compounds and alcohols, though this peak was less prominent in the cigarette butt extract, possibly due to loss of hydration. Peaks detected in the region of 1550–1650 cm$^{-1}$ indicated the presence of aliphatic secondary amines, while a notable band near 2159 cm$^{-1}$ corresponded to the nitrile (C≡N) group, commonly attributed to alkaloid compounds such as nicotine and related pyridine alkaloids abundant in tobacco and cigarette butt extracts. The fingerprint region, spanning 1418–1699 cm$^{-1}$, contained several peaks, including C–H bending vibrations indicative of aliphatic hydrocarbons in both the extracts. Minor shifts observed in certain peaks of the cigarette butt extract compared to the tobacco leaf extract may reflect chemical environment changes resulting from the smoking process and decomposition. Overall, these FTIR results confirm the presence of key bioactive functional groups— hydroxyl, amine, and nitrile—highlighting the phytochemical complexity that contributes to the larvicidal activity observed in both extracts.

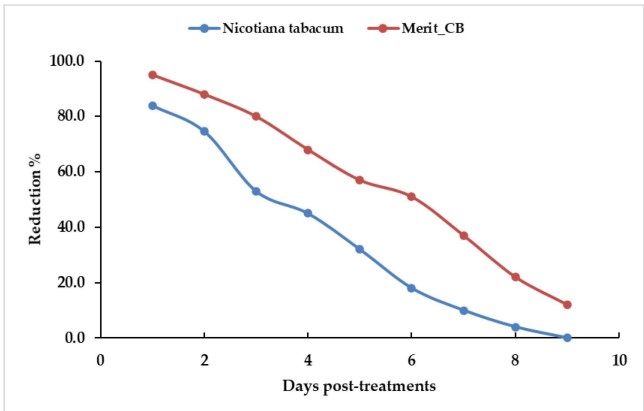

**Fig 8. Field evaluation for larvicidal efficacy of *Nicotiana tabacum* leaves (a) and Merit CB (b) methanol extracts treated at a dose of LC$_{95}$ X2 (194.2 and 30.9 ppm) in mosquito larval pools.**

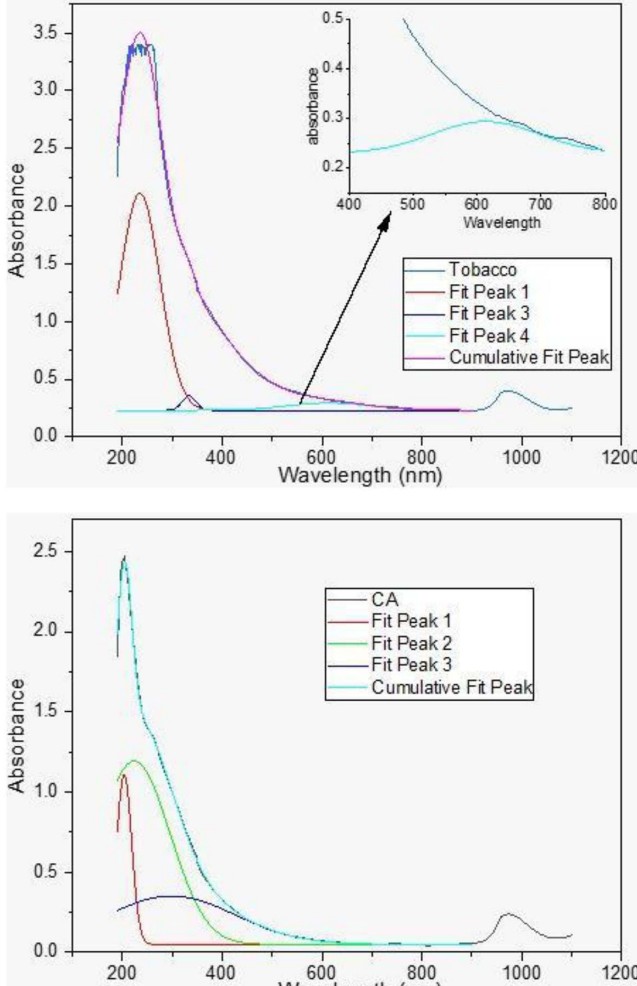

**Fig 9. Absorption of tobacco leaves (a) and Merit (b) cigarette but methanol extracts as function of wavelengths.**

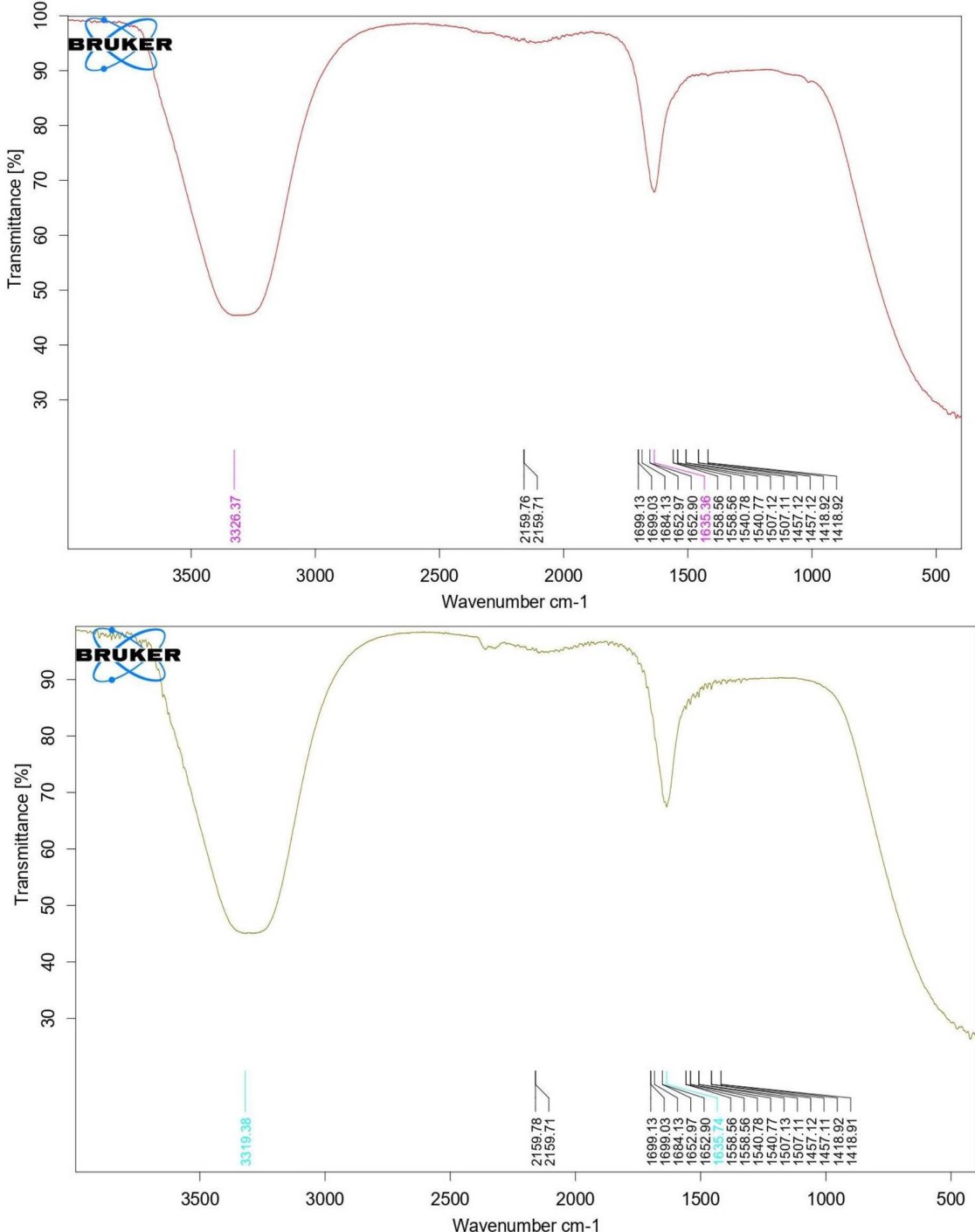

**Fig 10. FTIR plot of tobacco leaves (a) and Merit (b) cigarette butt methanol extract.**

**3.6.3 GC-MS data analysis.** Metabolomic analysis of the eight extracts and comparison between methanolic and aqueous extracts were done with the help of GC-MS analysis. Results of GC-MS analysis in our study led to the identification of various compounds such as terpenes, fatty acids, esters, ketone, alkane, steroids, aliphatic amine, and phenols in the leaves of *N. tabacum* and cigarette butt by using two different solvents (methanol and aqueous) (Tables 9–11). *N. tabacum* methanol leaf extract contained 13 compounds (Table 9), while the aqueous leaf extract contained two compounds, in which Pyridine, 3-(1-methyl-2-pyrrolidinyl)-, (S)- (86.21%), was more abundant in both the extracts; 3,7,11,15-Tetramethyl-2-hexadecen-1-ol (4.73%) and Phytol (2.03%) in the methanol extract, while Pyridine, 3-(1-methyl-2-pyrrolidinyl)-, (S)- (98.5%),

**Table 9. The major chemical constituents in *Nicotiana tabacum* methanol extracts.**

| No | RT | Compound Name | Area % | M. F. | M.W | Chemical class |
|----|----|---------------|--------|-------|-----|----------------|
| 1 | 4.62 | Pentanoic acid, 3-methyl- | 1.18 | $C_6H_{12}O_2$ | 116 | methyl fatty acids |
| 2 | 10.32 | Pyridine, 3-(1-methyl-2-pyrrolidinyl)-, (S)- | 86.21 | $C_{10}H_{14}N_2$ | 162 | pyridine alkaloid |
| 3 | 19.26 | 3,7,11,15-Tetramethyl-2-hexadecen-1-ol | 4.73 | $C_{20}H_{40}O$ | 296 | diterpene |
| 4 | 20.76 | Hexadecanoic acid, methyl ester | 0.26 | $C_{17}H_{34}O_2$ | 270 | methyl palmitate |
| 5 | 22.17 | Quercetin | 0.84 | $C15H_{10}O_7$ | 302 | phenol |
| 6 | 23.40 | 9,12-Octadecadienoic acid, methyl ester, (E,E)- | 1.22 | $C_{19}H_{34}O_2$ | 294 | fatty acid |
| 7 | 23.80 | Phytol | 2.03 | $C_{20}H_{40}O$ | 296 | diterpene |
| 8 | 29.09 | 1-Heptatriacotanol | 1.14 | $C_{37}H_{76}O$ | 536 | fatty acid |
| 9 | 24.82 | 9,12,15-Octadecatrienoic acid, (Z,Z,Z)- | 0.43 | $C_{18}H_{30}O_2$ | 278 | linolenic Acid |
| 10 | 27.25 | 1-methyl-5S-(3-pyridinyl)-2-pyrrolidinone | 0.58 | $C_{10}H_{12}N_2O_2$ | 176 | cotinine |
| 11 | 31.20 | Dotriacontane | 0.82 | $C_{32}H_{66}$ | 450 | long-chain alkane |
| 12 | 34.96 | Ethyl iso-allocholate | 0.56 | $C_{26}H_{44}O_5$ | 436 | steroid |

**Table 10. The major chemical constituents in *Nicotiana tabacum* aqueous extracts.**

| No | RT | Compound Name | Area % | M. F. | M.W | Chemical class |
|----|----|---------------|--------|-------|-----|----------------|
| 1 | 10.16 | Pyridine, 3-(1-methyl-2-pyrrolidinyl)-, (S)- | 98.50 | $C_{10}H_{14}N_2$ | 162 | pyridine alkaloid |
| 8 | 31.52 | Docosane | 1.50 | $C_{22}H_{46}$ | 310 | alkane |

**Table 11. The major chemical constituents in Merit cigarette butt methanol extracts.**

| No | RT | Compound Name | Area % | M. F. | M.W | Chemical class |
|----|----|---------------|--------|-------|-----|----------------|
| 1 | 7.44 | Tetradecane, 1-chloro- | 1.52 | $C_{14}H_{29}Cl$ | 232 | myristyl chloride |
| 2 | 13.13 | Pyridine, 3-(1-methyl-2-pyrrolidinyl)-, (S)- | 71.18 | $C_{10}H_{14}N_2$ | 162 | pyridine alkaloid |
| 3 | 26.37 | Cholestan-3-ol, 2-methylene-, (3á,5à)- | 0.97 | $C_{28}H_{48}O$ | 400 | wax |
| 4 | 27.46 | Hexadecanoic acid, 2,3-dihydroxypropyl ester | 5.23 | $C_{19}H_{38}O_4$ | 330 | suberic acid |
| 5 | 27.66 | Estra-1,3,5(10)-trien-17á-ol | 1.33 | $C_{18}H_{24}O$ | 256 | phenol |
| 6 | 27.80 | Phytol | 1.03 | $C_{20}H_{40}O$ | 296 | diterpene |
| 7 | 30.67 | 9-Octadecenoic acid (z)- | 6.80 | $C_{18}H_{34}O_2$ | 282 | oleic acid |
| 8 | 30.83 | Hi-oleic safflower oil | 0.67 | $C_{21}H_{22}O_{11}$ | 450 | fatty acids |
| 9 | 32.67 | 1-methyl-5S-(3-pyridinyl)-2-pyrrolidinone | 0.58 | $C_{10}H_{12}N_2O_2$ | 176 | cotinine |
| 10 | 31.09 | 1-Heptatriacotanol | 4.26 | $C_{37}H_{76}O$ | 536 | fatty acid |
| 11 | 31.20 | Dotriacontane | 0.82 | $C_{32}H_{66}$ | 450 | long-chain alkane |
| 12 | 32.00 | 1,25-Dihydroxyvitamin D3, TMS derivative | 2.62 | $C_{30}H_{52}O_3Si$ | 488 | vitamin D3 |
| 13 | 36.42 | Ethyl iso-allocholate | 2.99 | $C_{26}H_{44}O_5$ | 436 | steroid |

Pyridine alkaloid OR Nicotine.

and Docosane (1.5%) in the aqueous extract (Table 10). Merit cigarette butt methanol extract also had 13 compounds, as shown in Table 11. The highest concentrations of the detected compounds were Pyridine, 3-(1-methyl-2-pyrrolidinyl)-, (S)-(71.18%), 9-Octadecenoic acid (z)- (6.80%), and Hexadecanoic acid, 2,3-dihydroxypropyl ester (5.23%).

## 5. Discussion

Plant extracts and essential oils contain a number of important natural components that are safe to use in pest and disease control because they effectively eliminate dangerous pests and then degrade naturally in the environment [33]. Biopesticides account for only 5% of pesticides worldwide, despite their advantages as insecticides [34]. However, due to their special properties that enhance their application, such as their non-toxicity to the environment, the use of biopesticides represents a distinct solution from chemical pesticides. People use different species of *Nicotiana*, which grow naturally around the world, for both recreational and medicinal purposes. More than 120 nations grow it due to its high economic value [35]. Despite the identification of more than 75 tobacco species, only *N. rustica* L. and *N. tabacum* L. are currently grown commercially for human use [36].

Cigarette butt wastes, containing nicotine and other compounds, causing harm to our environment, can be a low-cost and easy-to-produce alternative to traditional pesticides. They do not impact environmental adversely due to their rapid biodegradation and low accumulation in the environment, making them a more sustainable and environmentally friendly option in mosquito control programs. The present studies evaluated the comparative mosquito control potential of *N. tabacum* leaves and cigarette butt wastes extracted in methanol and water. Our data indicated that CB extracts had a significant killing effect on both *Cx. pipiens* and *Ae. aegypti* larvae compared to tobacco leaves extracts. Additionally, the study indicated that methanolic extracts of tobacco leaves and cigarette butts were more effective at killing mosquito larvae than aqueous extracts. The quality or brands of cigarette butts also played a role in efficacy of killing mosquito larvae as observed in mortality% induced imported species represented by Merit and LM (100 MO%) compared to the local brands Cleopatra and Boston (90, 85 MO%). This variation in larvicidal potential may be attributed to the quality of the tobacco and the chemical contents it contains. A similar study evaluated the efficacy of CB Marlboro extract against *Ae. aegypti* larvae in their 2nd and 4th instars. The CB solutions (2CBSol and 3CBSol) were found to be highly effective in killing *Ae. aegypti* [29]. It is suggested that tobacco, along with some other industrial elements, contributes to the toxicity of cigarette butts. Many studies have looked at how tobacco plant extracts act on different types of pests, medical as well as agricultural. For example, Ileke, Oyeniyi [37] revealed how tobacco leaf and seed extracts affected *Anopheles gambiae* larvae, pupae, and adults. They found that both the extracts of *N. tabacum* could kill all the treated larvae, pupae, and adults of the *An. gambiae* mosquito at the highest concentration (0.5%) used in the experiment. The $LC_{50}$ results indicated that the leaf extract showed higher toxicity than the seed extract against different developmental stages. The $LC_{50}$ values for the extracts of *N. tobacco* leaves and seeds against the larvae were 0.153 µg/mL and 0.188 µg/mL, respectively, while the corresponding values for adults were 0.219 µg/mL and 0.290 µg/mL, indicating comparatively lower toxicity of extracts against adults. Kanmani, Kumar (38) demonstrated the harmful effects of *N. tabacum* leaf extracts on rice weevil-stored products, *Sitophilus oryzae*. They reported that the chloroform and acetone extracts from tobacco leaves possessed the higher insecticidal activity against adult weevils in comparison to hexane, ethyl acetate, petroleum ether, dichloromethane, methanol, and aqueous extracts. Some of the active compounds in tobacco extract that may be responsible for the effects were suggested to be nicotine, nicotinic acid, cotinine, indole, farnesol, sclareol, squalene, and palmitic acid.

In another study, Lamria, Gozan [38] investigated the potential of using tobacco leaf extract as an effective insecticidal agent for smoke production and fogging against *Ae. aegypti.* They used propylene glycol and glycerol as solvents instead of petroleum fuel. The toxicity test results using the pyrolysis extract indicated that concentrations of 10%–20% did not achieve $LC_{50}$, while 30% concentration reached $LC_{50}$ within 9 min. The 40% concentration reached $LC_{50}$ at the 6th min, while the 50% extract achieved $LC_{50}$ at the 5th min. A study by Hegazy and Baz [39] looked at the toxic effects of five types of wastes—cigarette filters, apricot kernels, Lipton packet tea, date palm kernels, and burnt rice straw— on the third stage

of the *Cx. pipiens* insect larvae in a test tube. They also investigated their effects on the number of eggs laid and the number of eggs that hatched. They found that all plant and industrial waste extracts exhibited appreciable larvicidal activity, with cigarette filters achieving 100% mortality and Lipton packet tea had 97% effectiveness which was followed by date palm kernels at 90%, burnt rice straw at 84%, and apricot kernel extracts at 79%. The corresponding $LC_{50}$ values obtained were 528.14, 582.42, 815.5, 976, and 1080.50 ppm after 24 hours, respectively.

*Nicotiana* species is known to have many phytochemical metabolites, such as diterpene glycosides and acyclic hydroxygeranyl linalool. Not only does nicotine break down into metabolites, but so do isoprenoids, aromatic compounds, flavonoids, cembranoid compounds, volatiles, and pyridine alkaloids [40]. Moreover, these molecules may have crucial biological functions. Researchers have identified thousands of metabolites from tobacco plants, both chemically and physiologically, and continue to investigate their chemical composition [41].

Nevertheless, an adequate understanding of mosquito behavior and biology is significant since it can aid in the prevention of numerous diseases they may transmit. Water bodies serve as the primary sites for most female mosquitoes to lay their eggs, leading to the development of the mosquito larvae. Therefore, natural or artificial water sources influence mosquito abundance and prevalence [42]. Many studies establish a relationship between water quality and mosquito abundance as well as prevalence. As an example, mosquito larvae and pupae survive maximally in water with a neutral pH [43]. Their survival rates drop below pH levels of 6–8 [44]. Yamada, Maiga [45] found the importance of dissolved oxygen along with food for oviposition and immature nesting of mosquitoes.

Other studies show different levels and directions of associations between water quality attributes mosquito number and presence [46]. These include mosquitoes' attraction to the color of the water medium, the turbidity, or the container color. Researchers have conducted studies on the reactions of mosquitoes to various plant colors in both field and laboratory settings. Researchers are particularly interested in plants that contain color or pigments that attract mosquitoes, such as *Hibiscus sabdariffa, Bougainvillea glabra, Acacia indigo, Delonix regia*, and other plant colors [30,47].

Hence, the cigarette butt extract was evaluated to attract female mosquitoes to lay eggs, and it was found that the cigarette butt extract has a reddish-brown color that decreases or increases according to the concentration. The study found that all the Cleopatra, Merit, Boston, and LM CB extracts were appealing to female mosquitoes. The dark-colored cups (10 and 15 CBSO) attracted the gravid female mosquitoes at the greatest number to lay a high number of eggs followed by cups with lighter concentrations (5 CBSO) or those without any treatment. Furthermore, the dark-colored cups had a lower egg hatching rate than the light-colored or untreated cups. This phenomenon proved useful in our current study, as the colors of the cups attracted female mosquitoes to lay their eggs, while the extract either killed the larvae or penetrated the egg, resulting in the embryo's death. In a study that ran concurrently with the current work, Baz, Alfagham (30) discovered that female *Cx. pipiens* laid about 65 egg rafts at a high concentration of *Acacia nilotica* extract (1000 ppm) with a hatching rate of 24.4%, while at moderate and low concentrations (500 and 250 ppm), they laid a moderate number of egg rafts (37 and 15 eggs) with a hatching rate of 48.4% and 76.5%, respectively.

Visually-directed mosquito activities are believed to contribute to host attraction. Researchers have long established that dark, high-contrast items attract mosquitoes more, leading to the formation of black traps [48,49]. Recent research indicates that specific scents enhance visual search activities in *Ae. aegypti* mosquitoes when presented with high-contrast visual cues. Alonso San Alberto, Rusch [50] found that vision is crucial for biting insects, including mosquitoes, tsetse flies, and kissing bugs since it facilitates flight control, object tracking for host or nectar acquisition, and the identification of oviposition locations. Other host-related signals, such as odor and temperature, intrinsically link to the visual inputs that facilitate these activities.

Previous research suggests that egg-laying signals are primarily visual, revealing a preference for laying eggs in dark containers and dark-colored water [51,52]. According to De Obaldia, Morita [53] mosquitoes attract their prey not only through the presence of carbon dioxide and heat but also through visual attraction. They do not perceive colors the same way we do. They have eyes that detect shapes, objects, and colors in detail. According to recent research, mosquitoes

exhibit strong attraction to specific colors, specifically orange, amber, and red, within the wavelength range of 510–660 nm [50]. This preference likely helps them locate potential hosts for breeding or oviposition. Thus, their reactions to long wavelengths (red to human observers) may indicate achromatic responses from visual channels attuned to medium wavelengths, which are consequently perceived as dark gray or black when contrasted with a light background. In our work, the primary distinction between the absorption profiles of tobacco and cigarette butt extracts is the existence of an additional peak in the visible region of tobacco absorption at 605 nm. It is suggested that this could give tobacco samples an advantage in luring adult female mosquitoes for oviposition [54]. Mosquito eyes are believed to be susceptible to seeing or flying onto specific colors and wavelengths; for instance, female mosquitoes are subjected to strongly preferring objects at wavelengths 600 nm and 660 nm [50].

The present study demonstrated that *N. tabacum* leaves and cigarette butt extracts decreased both larval mosquito populations in the field. Our data revealed that the Merit CB extract significantly reduced the number of larval mosquitoes in pools (95%) 24 h after treatment and continued to do so for six days. On the other hand, the *N. tabacum* leaves extract was only able to reduce the number of mosquito larvae 84% of the time.

A number of field studies have been conducted to test plant extracts or essential oils against mosquito larvae [39], Chaiyasit, Choochote [55]. A field study found that fennel oil and NLC-F killed 89.8% and 97.4% of the mosquito larvae 24 h after treatment, while green tea oil and NLC-GT killed 89% and 93% of the larvae at the same time [56]. NLC-F stayed effective for 8 days, and NLC-GT stayed effective for 7 days. Some studies have confirmed the effectiveness of using tobacco waste on mosquito larvae and a number of insects and acari [54].

Secondary metabolites such as alkaloids, terpenoids, flavonoids, saponins, tannins, and carbohydrates give medicinal plants their pharmacological effects. Botanical insecticides contain a wide range of biologically active substances that can harm pests or organisms that consume or contact these [57]. Tobacco extracts and cigarette butts may exhibit diverse biological activities on insects. Our study shows that of all the extracts evaluated—both tobacco leaves extracts and industrial tobacco methanol extracts represented by cigarette butts—the cigarette butt extracts of the Merit brand caused the highest mortality rate in the larvae of *Cx. pipiens* and *Ae. aegypti*. It has been reported that tobacco in general contains pyridine and diterpene alkaloids, along with phenols [58]. It is known that these two groups of chemicals have many biological effects, such as being effective against vector insects [59,60], which may help explain the effects that these extracts are said to have.

This is in agreement with our results, as the GC-MS analysis showed the maximum abundance of alkaloids (pyridine alkaloids) among the chemical biological compounds in tobacco and all brands of cigarette butt extracts. Terpenes, fatty acids, phenols, alkenes, and other chemical constituents followed closely behind. Zhang, Pan [61] have indicated that recent medical and pharmaceutical plants have demonstrated the unique and plentiful functional metabolites present in tobacco, such as nicotine, solanesol, cimpranoid diterpenes, essential oil, and seed oil. These metabolites, which circumvent the toxic constituents of smoke, primarily exhibit antioxidant, anti-lipid production, pro-lipid oxidation, pro-insulin sensitivity, anti-inflammatory, anti-apoptotic, and antimicrobial properties.

Alkaloids constitute the most significant category of natural compounds, serving a crucial function in insecticidal applications [21]. Wachira, Omar [59] concluded that pyridine alkaloids derived from *Ricinus communis* are effective against the malaria vector *An. gambiae*. It was found that the furocoumarin and quinolone alkaloids in *Ruta chalepensis* leaves stopped *Spodoptera littoralis* larvae from feeding and killed them [62]. Acheuk and Doumandji-Mitiche [63] discovered that the alkaloid extract of *Pergularia tomentosa* exhibited antifeeding and larvicidal properties. Lee [64] determined that piperine and piperidine alkaloids exhibit larvicidal efficacy against mosquitoes. Some chemicals found in *Arachis hypogaea* extract can kill young mosquitoes that transmit chikungunya and malaria [65].

*Aspidosperma rigidum* air-dried, and ground leaves and bark contained capoxin A. It was highly effective in killing parasites at a concentration of 100 µg/mL. It was more harmful to *Leishmania infantum* than to *T. cruzi*, killing 82.13% and 69.92%, respectively. Capoxin B showed a 68.92% mortality rate against *T. cruzi*, while carabanopin was inactive. All

of these compounds showed no toxicity toward mammalian CHO cells [66]. From our perspective, utilizing cigarette butt waste as a source for bio-insecticides represents a promising and sustainable approach for mosquito larval control. Cigarette butts are abundantly available and constitute a significant environmental burden. Repurposing them for mosquito management would not only address environmental pollution but also enhance the ecological and economic efficiency of pest control strategies. Our results demonstrated that methanolic extracts of certain cigarette butt brands, especially Merit, showed potent larvicidal activity against *Cx. pipiens* and *Ae. aegypti*, with minimal effects on non-target organisms. Additionally, production from such waste could offer a cost-effective and straightforward alternative to conventional chemical insecticides. However, there are challenges associated with variability in chemical composition across different cigarette brands and concerns about the presence of heavy metals and toxic compounds in cigarette butts. This necessitates careful evaluation of environmental safety and the development of efficient and safe extraction methods to ensure consistent efficacy and minimize potential adverse effects. Therefore, while cigarette butt-derived bio-insecticides have enormous potential, further research is essential to validate their safety, optimize extraction, and assess their performance in large-scale field applications before widespread adoption as a sustainable alternative.

## Conclusion

Mosquitoes are significant vectors of deadly diseases that pose ongoing threats to human health, spreading across endemic, non-endemic, and new regions worldwide. While synthetic insecticides are commonly used to curb mosquito-borne diseases, increasing resistance among mosquito populations and concerns over environmental and non-target organism toxicity have driven the demand for safer, biodegradable, and more effective control alternatives. Phytochemicals have gained attention in mosquito control due to their natural origin, environmental safety, lower toxicity, cost-effectiveness, and reduced likelihood of inducing resistance. In this study, extracts from tobacco plants (*N. tabacum* L.) and cigarette butt waste were evaluated and demonstrated clear larvicidal efficacy against *Cx. pipiens* and *Ae. aegypti* mosquito larvae, with methanol extracts showing higher potency than aqueous ones. Certain cigarette butt brands, notably Merit, exhibited particularly strong insecticidal activity, suggesting their potential as natural alternatives to conventional chemical insecticides. The extracts showed limited adverse effects on non-target organisms, supporting their safe integration into mosquito management programs. Chemical analyses identified alkaloids (nicotine and related pyridine alkaloids), flavonoids, and terpenes as key active components responsible for this insecticidal activity. Future work should focus on conducting thorough environmental safety assessments, particularly regarding the concentrations of heavy metals and semi-toxic compounds in cigarette butt extracts, as well as on developing safer alternatives. Additionally, detailed studies on the mechanisms of action and the potential for combining these extracts within multifaceted integrated pest management strategies are essential to enhance sustainability and mitigate resistance development.

## Author contributions

**Conceptualization:** Mohamed M. Baz, Yasser A. El-Sayed, Mohammed E. Gad, Sarita Kumar, Abdelfattah Selim, Abeer M. Alkhaibari.

**Data curation:** Mohamed M. Baz, Yasser A. El-Sayed, Mohammed E. Gad, Mohammed H. Alruhaili, Hattan S. Gattan, Reham M. Mostafa, Sarita Kumar, Ibrahim E. Hussein, Abdelfattah Selim, Abeer M. Alkhaibari.

**Formal analysis:** Yasser A. El-Sayed, Samah Abdel Gawad, Mohammed E. Gad, Moustafa Ibrahim, Mohammed H. Alruhaili, Hattan S. Gattan, Reham M. Mostafa, Ibrahim E. Hussein, Abdelfattah Selim.

**Funding acquisition:** Mohamed M. Baz, Yasser A. El-Sayed, Samah Abdel Gawad, Mohammed E. Gad, Mohammed H. Alruhaili, Hattan S. Gattan, Sarita Kumar, Abdelfattah Selim, Abeer M. Alkhaibari.

**Investigation:** Mohamed M. Baz, Yasser A. El-Sayed, Mohammed E. Gad, Moustafa Ibrahim, Mohammed H. Alruhaili, Reham M. Mostafa, Abdelfattah Selim, Abeer M. Alkhaibari, Ahlam F. Moharam.

**Methodology:** Mohamed M. Baz, Yasser A. El-Sayed, Samah Abdel Gawad, Moustafa Ibrahim, Hattan S. Gattan, Ibrahim E. Hussein, Abdelfattah Selim, Abeer M. Alkhaibari, Ahlam F. Moharam.

**Project administration:** Mohammed H. Alruhaili.

**Software:** Mohammed E. Gad, Mohammed H. Alruhaili, Ahlam F. Moharam.

**Supervision:** Mohamed M. Baz, Mohammed E. Gad, Abeer M. Alkhaibari.

**Validation:** Mohamed M. Baz, Sarita Kumar, Ibrahim E. Hussein.

**Visualization:** Yasser A. El-Sayed, Samah Abdel Gawad, Moustafa Ibrahim, Hattan S. Gattan, Reham M. Mostafa, Sarita Kumar, Ibrahim E. Hussein.

**Writing – original draft:** Mohamed M. Baz, Yasser A. El-Sayed, Samah Abdel Gawad, Mohammed E. Gad, Moustafa Ibrahim, Mohammed H. Alruhaili, Hattan S. Gattan, Reham M. Mostafa, Sarita Kumar, Abdelfattah Selim, Abeer M. Alkhaibari, Ahlam F. Moharam.

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
