## [Decision Letter · Decision Letter 0]

5 Aug 2025

PONE-D-25-33977Tobacco plant and cigarette butt wastes: field and laboratory evaluation of insecticidal activities against Culex pipiens, Aedes aegypti (Diptera: Culicidae), and non-target organismsPLOS ONE

Dear Dr. Selim,

Thank you for submitting your manuscript to PLOS ONE. After careful consideration, we feel that it has merit but does not fully meet PLOS ONE’s publication criteria as it currently stands. Therefore, we invite you to submit a revised version of the manuscript that addresses the points raised during the review process.

**ACADEMIC EDITOR NOTES: **

Dear authors, all the referees raised important questions about the manuscript. I suggest that the authors conduct a thorough review of the manuscript; the response letter should contain a point-by-point answer to all the referees' questions. Furthermore, all changes to the text should be highlighted in yellow.

We look forward to receiving your revised manuscript.

Kind regards,

Jorddy Neves Cruz

Academic Editor

PLOS ONE

3. In the online submission form, you indicated that [data available on request form corresponding author].

Additional Editor Comments (if provided):

Reviewers' comments:

Reviewer's Responses to Questions

**Comments to the Author**

1. Is the manuscript technically sound, and do the data support the conclusions?

Reviewer #1: No

Reviewer #2: Yes

Reviewer #3: Yes

2. Has the statistical analysis been performed appropriately and rigorously?

Reviewer #1: Yes

Reviewer #2: Yes

Reviewer #3: Yes

3. Have the authors made all data underlying the findings in their manuscript fully available?

Reviewer #1: Yes

Reviewer #2: Yes

Reviewer #3: Yes

4. Is the manuscript presented in an intelligible fashion and written in standard English?

Reviewer #1: No

Reviewer #2: Yes

Reviewer #3: Yes

5. Review Comments to the Author

Reviewer #1: Comments to the authors

The manuscript titled “Tobacco plant and cigarette butt wastes: field and laboratory evaluation of insecticidal activities against Culex pipiens, Aedes aegypti (Diptera: Culicidae), and non-target organisms”. The present study evaluated about to show that certain chemicals in the tobacco plant (Nicotiana 34 tabacum) and cigarette butt wastes (CBWs) are crucial in biological processes against mosquitoes. After carefully reviewing this manuscript I can not see merit in this research and also the current form contains several writing, scientific, grammatical and typographical errors so please carefully correct the errors based on the following comments;

1. The manuscript title is hard to follow, rewrite the title.

2. Line 26-33: scientifically not sound.

3. Line 33-37:the objective and methodology not clear.

4. The end of the abstract the conclusion is missing.

5. If possible add a few more keywords and avoid the keyword from the manuscript title.

6. Line 51-83: scientifically not sound, the current form looks like a textbook. Please carefully revise it.

7. Line 87: references are missing.

8. Line 91: add a few more recent references.

9. Line 91-92: hard to follow.

10. The entire manuscript is scientifically not sound and hard to follow.

11. In the introduction the rationale of present study is not properly described.

12. Provide latitude and longitude information of plant sample collected areas.

13. The plant sample authenticated by any plant taxonomist? If yes, provide the details. If not then how did you identify the plant?

14. Line 132-133: provide the room temperature and humidity details.

15. Line 137: why methanol solvent? What about other solvents?

16. Did you extract the Cigarette butt? Or you extracted Cigarette? If you extracted Cigarettes, how do you recycle the Cigarette butt? I think the author is confused with this research.

17. Line 196: what is the nutrient content of Tetramin?

18. Line 196-197: not clear.

19. Line 203: typo error.

20. The methodological part is poorly written, the current form of the section scientifically not sound. So revise this part with proper references, I do not see any proffer reference in the methods section.

21. The results part must be improve

22. In the discussion part scientifically not sound and don’t provide your results in the discussion part. This is not the right place to write your results.

23. The reference section contains several formatting and typographical errors so carefully correct the errors.

24. Throughout the manuscript I can see several scientific, technical, fundamental, grammatical and typographical errors so please carefully correct.

Reviewer #2: The manuscript entitled “Tobacco plant and cigarette butt wastes: field and laboratory evaluation of insecticidal activities against Culex pipiens, Aedes aegypti (Diptera: Culicidae), and non-target organisms” by Baz et al. is well-written in simple and clear English and presents valuable information.

While reading through the manuscript, I found it easy to understand due to the straightforward language used. Below are a few comments from my side:

1. All scientific names must be italicized and given in full at first mention (e.g., Aedes aegypti Linnaeus). Afterwards, abbreviate genus (e.g., Ae. aegypti). In the title, include the author citation (“Say”) after Aedes aegypti.

2. Abbreviations should be expanded upon their first use.

3. Please provide the author citation (taxonomic authority) for the mosquito species in the title (e.g., “Say”).

4. The Introduction is well-written but could be enhanced by including more detailed information about the tobacco plant, including nicotine and other compounds it contains.

5. The Introduction would benefit from more detail on tobacco’s alkaloids (nicotine, nornicotine, anatabine), flavonoids and terpenes, including approximate concentrations in Nicotiana tabacum L. leaves.

6. Correct the repeated name in line 112.

7. Cigarette butts contain many toxic compounds, such as arsenic, a heavy metal. Highlighting the insecticidal potential of cigarette butt-derived products could strengthen the manuscript. "What about the toxicity of the present compounds? Is it reliable and safe to use them as insecticides?".

8. Add a sentence in the Introduction (or early Discussion) succinctly stating how cigarette‑butt–derived larvicides compare in cost, environmental impact, or ease of production versus conventional options.

9. The Materials and Methods and Results sections are well-structured and clearly presented.

10. Was any standard control (Pesticide) used for comparison in this study? Please clarify.

11. Figures 4 to 8 currently show mean values only. Add standard error or confidence interval bars to all bar charts and line plots to indicate variability.

12. Was ethical clearance obtained for the use of hamster mouse in this study? If so, please provide the relevant details in the section 2.3.1.

13. The Discussion section is well-structured but is overly lengthy. Consider condensing or reorganizing it for better flow and add the limitation and future direction of the study.

14. The Conclusion should be rewritten. Please summarize the core findings and include future perspectives.

15. Cross‑check that all figures/tables are cited in order.

16. The entire manuscript should be thoroughly checked for grammatical, typographical, and spacing errors.

Reviewer #3: 1. Lines 26 to 33 of the abstract should be rewritten in more concise and informative way.

2. The abstract lacks the main phytochemical results. Authors should highlight the main identified phytoconstituents.

3. In the plant collection part, kindly add the name of the botanist who authenticated the collected tobacco plant.

4. In lines 182 and 187, why only the plant extract and one brand were selected for analysis rather than the other collected cigarette butt samples?

5. Upon collecting the cigarette butts from public areas, how could the authors differentiate the exact brand of each of them?

6. The authors are advised to check for any Grammatik or spelling mistakes throughout the manuscript.

7. Line 352, the authors mistype “acetone extract” although the study is for methanol extract. correct please.

8. A standardization method, using HPLC for example, for the percentage of different phytochemicals present in the cigarette butt extracts is requested.

9. The discussion part is too long and need to be rewritten in more informative way.

10. The discussion part may include the authors’ opinion about the feasibility of using cigarette butt waste as a source for bio insecticide.

11. In the references part, the latin names should be italic.

12. In table 9, instead of bioactive compounds in the header, the authors may use chemical class or class.

13. In fig. 2 kindly add the scale used in capturing this picture.

14. Kindly provide an explanation for figure 10, the FTIR results, in the results section.

6. PLOS authors have the option to publish the peer review history of their article (what does this mean?). If published, this will include your full peer review and any attached files.

Reviewer #1: No

Reviewer #2: No

Reviewer #3: No

---

## [Author Response · Author response to Decision Letter 1]

27 Aug 2025

Respected Prof. Dr. …………

Thanks to the Editorial board and reviewers for their careful and valuable review, including comments, opinions, directions, and their kind words about the manuscript.

• More recent information was added to the text sections.

• All manuscript sections were improved as recommended.

• Corrections were highlighted in yellow color.

We appreciate and value the effort you put into our manuscript to make it satisfactory and high quality enough for publication in your respected journal.

Reviewer #1: Comments to the authors

The manuscript titled “Tobacco plant and cigarette butt wastes: field and laboratory evaluation of insecticidal activities against Culex pipiens, Aedes aegypti (Diptera: Culicidae), and non-target organisms”. The present study evaluated about to show that certain chemicals in the tobacco plant (Nicotiana 34 tabacum) and cigarette butt wastes (CBWs) are crucial in biological processes against mosquitoes. After carefully reviewing this manuscript I can not see merit in this research and also the current form contains several writing, scientific, grammatical and typographical errors so please carefully correct the errors based on the following comments;

Response: Thank you for your careful review and constructive comments.

1. The manuscript title is hard to follow, rewrite the title.

Response: Checked and rewritten.

2. Line 26-33: scientifically not sound.

Response: Revised and corrected (Page 2, line 27-36).

3. Line 33-37: the objective and methodology not clear.

Response: Revised and corrected (Page 2, line 31-36).

4. The end of the abstract the conclusion is missing.

Response: Revised and added (Page 2, line 48-49).

5. If possible, add a few more keywords and avoid the keyword from the manuscript title.

Response: Revised and added (Page 2, line 50-51).

6. Line 51-83: scientifically not sound, the current form looks like a textbook. Please carefully revise it.

Response: Revised and Corrected (Page 3, line 53-70).

7. Line 87: references are missing.

Response: Revised and added (Page 4, line 88).

8. Line 91: add a few more recent references.

Response: Revised and added (Page 5, line 107).

9. Line 91-92: hard to follow

Response: Revised and corrected (Page 4, line 92-94).

10. The entire manuscript is scientifically not sound and hard to follow.

Response: Checked and revised.

11. In the introduction the rationale of present study is not properly described.

Response: Revised and added (Page 5, line 95-109).

12. Provide latitude and longitude information of plant sample collected areas.

Response: Revised and added (Page 60, line 139).

13. The plant sample authenticated by any plant taxonomist? If yes, provide the details. If not then how did you identify the plant?

Response: Revised and added (Page 6, line 143-144).

14. Line 132-133: provide the room temperature and humidity details.

Response: Revised and added (Page 6, line 144).

15. Line 137: why methanol solvent? What about other solvents?

Response: Methanol is used as a solvent for plant extractions because of its high polarity, which enables it to extract a wide range of bioactive plant compounds, such as alkaloids and phenols. It also effectively penetrates plant cell walls, increasing the yield of extracts. In addition to its easy evaporation and compatibility with analytical techniques. Furthermore, it does not require other solvents in our study for the variety of cigarette butts and tobacco plants. So, methanol is a practical and economical choice for laboratory research, as demonstrated in previous research

16. Did you extract the Cigarette butt? Or you extracted Cigarette? If you extracted Cigarettes, how do you recycle the Cigarette butt? I think the author is confused with this research.

Response: Sir, there is no confusion, as the remaining tobacco was extracted from the cigarette, represented by the cigarette butt, and this is what was explained in Figure 3.

17. Line 196: what is the nutrient content of Tetramin?

Response: Tetramin fish food is a commercial food commonly used in fish and aquatic worm farming. It is rich in proteins, vitamins, and minerals that support the growth of insects and fish, including mosquito larvae in laboratory experiments. Tetramin fish food is a commonly used compound in laboratories that breed mosquitoes for research purposes.

18. Line 196-197: not clear.

Response: Revised and corrected.

19. Line 203: typo error.

Response: Revised and corrected.

20. The methodological part is poorly written, the current form of the section scientifically not sound. So revise this part with proper references, I do not see any proffer reference in the methods section.

Response: Revised and corrected.

21. The results part must be improve

Response: Checked and corrected.

22. In the discussion part scientifically not sound and don’t provide your results in the discussion part. This is not the right place to write your results.

Response.: Discussion part has been revised as per the suggestions.

23. The reference section contains several formatting and typographical errors so carefully correct the errors.

Response: Care has been taken to remove errors.

24. Throughout the manuscript I can see several scientific, technical, fundamental, grammatical and typographical errors so please carefully correct.

Response: The manuscript has been carefully revised. Thanks for your suggestions.

Reviewer #2: The manuscript entitled “Tobacco plant and cigarette butt wastes: field and laboratory evaluation of insecticidal activities against Culex pipiens, Aedes aegypti (Diptera: Culicidae), and non-target organisms” by Baz et al. is well-written in simple and clear English and presents valuable information.

While reading through the manuscript, I found it easy to understand due to the straightforward language used. Below are a few comments from my side:

Response: Thank you for your careful review and constructive comments.

1. All scientific names must be italicized and given in full at first mention (e.g., Aedes aegypti Linnaeus). Afterwards, abbreviate genus (e.g., Ae. aegypti). In the title, include the author citation (“Say”) after Aedes aegypti.

Response: Revised and corrected. Sir, Carl Linnaeus first described Culex pipiens (Linnaeus) in 1758, while Aedes aegypti (Linnaeus) was described in 1762. Therefore, they are the original and formal scientific name given by Linnaeus.

2. Abbreviations should be expanded upon their first use.

Response: Checked and corrected.

3. Please provide the author citation (taxonomic authority) for the mosquito species in the title (e.g., “Say”).

Response: Revised and corrected.

4. The Introduction is well-written but could be enhanced by including more detailed information about the tobacco plant, including nicotine and other compounds it contains.

Response: Revised and added (Page 4, line 95-109).

5. The Introduction would benefit from more detail on tobacco’s alkaloids (nicotine, nornicotine, anatabine), flavonoids and terpenes, including approximate concentrations in Nicotiana tabacum L. leaves.

Response: Revised and added (Page 5, line 100-109).

6. Correct the repeated name in line 112.

Response: Checked and corrected.

7. Cigarette butts contain many toxic compounds, such as arsenic, a heavy metal. Highlighting the insecticidal potential of cigarette butt-derived products could strengthen the manuscript. "What about the toxicity of the present compounds? Is it reliable and safe to use them as insecticides?".

Response: We thank the reviewer for his important observation regarding the presence of toxic compounds in cigarette butt waste. Although cigarette butts contain toxic compounds such as arsenic, extracts obtained from plant waste (including tobacco waste) contain a variety of phytochemicals, including alkaloids (such as nicotine), flavonoids, terpenes, and other bioactive compounds with clear insecticidal effects. Furthermore, an assessment of the effect on non-target organisms showed that the extracts had negligible or statistically insignificant effects on most of these organisms, indicating the extracts' ability to target mosquito larvae without causing significant harm to beneficial organisms.

8. Add a sentence in the Introduction (or early Discussion) succinctly stating how cigarette butt–derived larvicides compare in cost, environmental impact, or ease of production versus conventional options.

Response: Revised and added in early discussion (Page 17, line 398-401).

9. The Materials and Methods and Results sections are well-structured and clearly presented.

Response: Thank you for your careful review and constructive comments.

10. Was any standard control (Pesticide) used for comparison in this study? Please clarify.

Response: In our current study, unfortunately, we did not use a standard control (pesticide), because there are many biological agents that are focused on the tobacco plants and their cigarette butt waste on two vector-borne diseases: Culex pipiens and Aedes aegypti.

11. Figures 4 to 8 currently show mean values only. Add standard error or confidence interval bars to all bar charts and line plots to indicate variability.

Response: Revised and added.

12. Was ethical clearance obtained for the use of hamster mouse in this study? If so, please provide the relevant details in the section 2.3.1.

Response: The research title, overall objective of the study, and ethical approval for the use of hamsters in this study are presented to the Ethics Committee of the Faculty of Science, and it is approved under Code (BUFS-REC-2025-355 Ent); (Page 10, line 223-226).

13. The Discussion section is well-structured but is overly lengthy. Consider condensing or reorganizing it for better flow and add the limitation and future direction of the study.

Response: Revised and Corrected.

14. The Conclusion should be rewritten. Please summarize the core findings and include future perspectives.

Response: Revised and rewritten.

15. Cross check that all figures/tables are cited in order.

Response: Checked and revised.

16. The entire manuscript should be thoroughly checked for grammatical, typographical, and spacing errors.

Response: Checked, revised and corrected.

Reviewer #3:

Response: Thank you for your careful review and constructive comments.

1. Lines 26 to 33 of the abstract should be rewritten in more concise and informative way.

Response: Revised and rewritten.

2. The abstract lacks the main phytochemical results. Authors should highlight the main identified phytoconstituents.

Response: Checked, added.

3. In the plant collection part, kindly add the name of the botanist who authenticated the collected tobacco plant.

Response: Revised and added.

4. In lines 182 and 187, why only the plant extract and one brand were selected for analysis rather than the other collected cigarette butt samples?

Response: Thank you for the important note. Tobacco plant extract and one cigarette butt brand (Cleopatra) were selected for spectroscopic analysis (UV-Visible and FTIR) because these samples represent the basic chemical composition and common active compounds found in most of the cigarette butt samples, we collected. The choice was indicated based on preliminary analyses that showed a high similarity in composition between the various cigarette butt samples, which led us to believe that analysing a single sample (CBs) with the tobacco plant (N. tabacum) was sufficient to provide a clear picture of the chemical constituents of this sample class (CBs) while saving the study resources and focusing on the results of interest. Accordingly, Cleopatra was selected as a common brand and representative of the chemical composition of other cigarette butt extracts.

5. Upon collecting the cigarette butts from public areas, how could the authors differentiate the exact brand of each of them?

Response: When collecting cigarette butts from public places, we relied on visual markers on filters and scrap paper to identify each cigarette pack's brand. We also relied on designs and logos printed on unfinished cigarette butts or filter pieces. Additionally, the brand identity of the collected cigarette butts was standardized and explained to individual staff working in these public places. This process helped reduce sample classification errors and ensure accurate analysis.

6. The authors are advised to check for any Grammatik or spelling mistakes throughout the manuscript.

Response: Checked, corrected.

7. Line 352, the authors mistype “acetone extract” although the study is for methanol extract. correct please.

Response: Revised and corrected (Methanolic), Page 16, line 378.

8. A standardization method, using HPLC for example, for the percentage of different phytochemicals present in the cigarette butt extracts is requested.

Response: Thank you for the important note. We would like to clarify that in the current study, HPLC was not applied or used to analyze and standardize the phytochemical ratios in cigarette butt extracts. Instead, GC-MS, FTIR, and UV-visible spectroscopy analyses were used to identify and characterize the active chemical compounds.

9. The discussion part is too long and need to be rewritten in more informative way.

Response: Checked and revised.

10. The discussion part may include the authors’ opinion about the feasibility of using cigarette butt waste as a source for bio insecticide.

Response: Revised and added (page 22, line 536-550).

11. In the references part, the latin names should be italic.

Response: Checked and revised.

12. In table 9, instead of bioactive compounds in the header, the authors may use chemical class or class.

Response: Revised and corrected in tables 9-11.

13. In fig. 2 kindly add the scale used in capturing this picture.

Response: Revised and Corrected.

14. Kindly provide an explanation for figure 10, the FTIR results, in the results section.

Response: Revised and added more results about the explanation of FTIR results.

---

## [Editor Report · Decision Letter 1]

3 Sep 2025

PONE-D-25-33977R1Field and Laboratory Assessment of Larvicidal Activity of Tobacco Plants and The Cigarette Butt Waste on Culex pipiens(Linnaeus, 1758), Aedes aegypti (Linnaeus, 1762) L. and Non-Target OrganismsPLOS ONE

Dear Dr. Selim,

Thank you for submitting your manuscript to PLOS ONE. After careful consideration, we feel that it has merit but does not fully meet PLOS ONE’s publication criteria as it currently stands. Therefore, we invite you to submit a revised version of the manuscript that addresses the points raised during the review process.

We look forward to receiving your revised manuscript.

Kind regards,

Jorddy Neves Cruz

Academic Editor

PLOS ONE

Journal Requirements:

1.If the reviewer comments include a recommendation to cite specific previously published works, please review and evaluate these publications to determine whether they are relevant and should be cited. There is no requirement to cite these works unless the editor has indicated otherwise. 

Additional Editor Comments:

Authors must provide a point-by-point review of all referees' queries. All modifications should be in yellow for better visualization of the manuscript changes.

---

## [Author Response · Author response to Decision Letter 2]

13 Sep 2025

Journal Requirements:

Point#: If the reviewer comments include a recommendation to cite specific previously published works, please review and evaluate these publications to determine whether they are relevant and should be cited. There is no requirement to cite these works unless the editor has indicated otherwise.

Response#; it edit

Additional Editor Comments:

Point#: Authors must provide a point-by-point review of all referees' queries. All modifications should be in yellow for better visualization of the manuscript changes.

Response#: all comments of reviewers highlighted by yellow color

---

## [Editor Report · Decision Letter 2]

16 Sep 2025

Field and Laboratory Assessment of Larvicidal Activity of Tobacco Plants and The Cigarette Butt Waste on Culex pipiens(Linnaeus, 1758), Aedes aegypti (Linnaeus, 1762) L. and Non-Target Organisms

PONE-D-25-33977R2

Dear Dr. Selim,

We’re pleased to inform you that your manuscript has been judged scientifically suitable for publication and will be formally accepted for publication once it meets all outstanding technical requirements.

Kind regards,

Jorddy Neves Cruz

Academic Editor

PLOS ONE
---

## [Editor Report · Acceptance letter]

PONE-D-25-33977R2

PLOS ONE

Dear Dr. Selim,

I'm pleased to inform you that your manuscript has been deemed suitable for publication in PLOS ONE. Congratulations! Your manuscript is now being handed over to our production team.

Kind regards,

on behalf of

Dr. Jorddy Neves Cruz

Academic Editor

PLOS ONE